# Analyzing the generalization capabilities of a hybrid hydrological model for extrapolation to extreme events

Eduardo Acuña Espinoza [1], Ralf Loritz [1], Frederik Kratzert [2], Daniel Klotz [2, 3], Martin Gauch [4], Manuel Álvarez Chaves [5], and Uwe Ehret [1]

[1]Institute of Water and Environment, Karlsruhe Institute of Technology (KIT), Karlsruhe, Germany
[2]Google Research, Vienna, Austria
[3]Helmholtz Centre for Environmental Research (UFZ), Leipzig, Germany
[4]Google Research, Zurich, Switzerland
[5]Stuttgart Center for Simulation Science, Statistical Model-Data Integration, University of Stuttgart, Stuttgart, Germany

**Correspondence:** Eduardo Acuña Espinoza (eduardo.espinoza@kit.edu)

**Abstract.** Data-driven techniques have shown the potential to outperform process-based models for rainfall-runoff simulation. Recently, hybrid models, which combine data-driven methods with process-based approaches, have been proposed to leverage the strengths of both methodologies, aiming to enhance simulation accuracy while maintaining certain interpretability. Expanding the set of test cases to evaluate hybrid models under different conditions, we test their generalization capabilities for extreme hydrological events, comparing their performance against Long Short-Term Memory (LSTM) networks and process-based models. Our results indicate that hybrid models show performance similar to that of the LSTM network for most cases. However, hybrid models reported slightly lower errors in the most extreme cases and were able to produce higher peak discharges.

## 1 Introduction

Data-driven techniques have demonstrated the potential to outperform process-based models for rainfall-runoff simulation (Kratzert et al., 2019; Lees et al., 2021; Feng et al., 2020). Moreover, Frame et al. (2022) addressed concerns about the generalization capability of data-driven methods for extrapolation to extreme events, demonstrating that Long Short-Term Memory (LSTM) networks (Hochreiter and Schmidhuber, 1997) outperformed process-based models in such scenarios.

Recently, hybrid models that combine process-based models with data-driven approaches have been proposed (Reichstein et al., 2019; Shen et al., 2023). The idea behind hybrid models is that they integrate the strengths of both process-based and data-driven approaches to improve simulation accuracy while maintaining a notion of interpretability (Jiang et al., 2020; Höge et al., 2022). Among the various approaches available to combine these methodologies, the parameterization of process-based models using data-driven techniques has shown promising results (Tsai et al., 2021). One way to interpret this technique is that it involves integrating a neural network with a process-based model in an end-to-end pipeline, where the neural network handles the parameterization of the process-based model. Alternatively, this can be viewed as a neural network with a process-based head layer, which not only compresses the information into a target signal but has a certain structure that allows for the

recovery of untrained variables. Kraft et al. (2022) applied this method, demonstrating that substituting poorly understood or challenging-to-parameterize processes with machine learning (ML) models can effectively reduce model biases and enhance local adaptivity. Similarly, Feng et al. (2022) and Acuña Espinoza et al. (2024) employed LSTM networks to estimate the parameters of process-based models, achieving state-of-the-art performance comparable with LSTMs and outperforming stand-alone conceptual models.

In a previous study, Acuña Espinoza et al. (2024) tested the performance and interpretability of hybrid models, with the overall goal of looking at the advantages provided by adding a process-based head layer to a data-driven method. They show that hybrid models can achieve comparable performance with LSTM networks, while maintaining a notion of interpretability. In this type of hybrid model, the interpretability arises from association. Specifically, the authors map the parameters and components of process-based models to predefined processes, domains, and states (e.g., baseflow, interflow, snow accumulation). While this approach does provide interpretability, it is important to clarify that this interpretability is indeed limited to associations, and may lack rigorous physical principles, especially when one uses models such as the Simple Hydrological Model (SHM) (Ehret et al., 2020) or the Hydrologiska Byråns Vattenbalansavdelning (HBV) (Bergström, 1992), which present significant simplifications of the underlying physical process. Moreover, Acuña Espinoza et al. (2024) also warn about the possibility that the data-driven section of the hybrid model compensates for structural deficiencies in the conceptual layer.

Building on this research line and expanding the set of test cases to evaluate hybrid models under different conditions, this study follows the procedure proposed by Frame et al. (2022) to investigate the ability of different models to predict out-of-sample conditions, focusing on their generalization capability to extreme events. We compare the performance of hybrid models against both traditional process-based models and stand-alone data-driven models. We aim to determine which model demonstrates higher predictive accuracy, particularly in simulating extreme hydrological events. We thereby address the following two research questions:

- How does a hybrid model compare to a process-based and a stand-alone data-driven model in the simulation of extreme hydrological events?

- Does hybrid modeling offer a higher performance than stand-alone data-driven approaches?

To achieve this objective, we have structured this article as follows: Section 2 describes the training/test data split and gives an overview of the different models. In Section 3, we compare the results of various tests that assess the generalization capabilities of data-driven, hybrid, and conceptual models. Lastly, Section 4 summarizes the key findings of the experiments and presents the conclusions of the study.

## 2 Data and methods

Donoho (2017) emphasize the importance of community benchmarks to drive model improvement. In the hydrological community, this practice has also been suggested to enable a fair comparison between new and existing methods (Shen et al., 2018; Nearing et al., 2021; Kratzert et al., 2024). Consequently, we built our experiments considering two existing studies. First,

we used the procedure proposed by Frame et al. (2022) to evaluate the generalization capability of different models (see section 2.1). In accordance with this study, the experiments were conducted using the CAMELS-US dataset (Addor et al., 2017; Newman et al., 2015), in the same subset of 531 basins. Second, we used the hybrid model architecture $\delta_n(\gamma^t, \beta^t)$, further explained in section 2.2.2, proposed by Feng et al. (2022). This architecture demonstrated competitive performance with LSTM networks in their original experiments, which also used the CAMELS-US dataset.

## 2.1 Data handling: training/test split

To produce an out-of-sample test dataset and evaluate the generalization capability of the different models to extreme streamflow events, we split the training and test set by years, based on the return period of the maximum annual discharge event. Closely following the procedure recommended by Bulletin 17C (England Jr et al., 2019), we fitted a Pearson III distribution to the annual maxima series of each basin, which we extracted from the observed CAMELS-US discharge records. We then calculated the magnitude of the discharge associated with different probabilities of exceedance. Using the discharge associated with the 5-year return period as a threshold, we classified the water years (here, a water year is defined as the period of time between the 1$^{st}$ of October and the 30$^{th}$ of September of the following year) into training or test set. Figure 1a shows an example of the training/test split for basin 01054200, in the northeast of the United States (US). The water years which contained only discharge records smaller than the associated 5-year threshold were used for training, while cases in which this threshold was exceeded were used for testing. It is important to note that there was a 365-day buffer between each training and testing period. The value of 365 days corresponds to the sequence length used by the LSTM model, and the buffer period avoids leaking test information during training. The results of the frequency analysis and the training/test data split for each basin can be found in the supplementary material accompanying this study. The original dataset contained 531 basins, each with 34 years of data (from 1980 to 2014), for a total of 18 054 years of data. After the data split process, 9 489 years were used for training, 3 429 for testing and 5 136 were buffers. Excluding the buffer data, 73% of the data was used for training and 27% for testing. This distribution is consistent with the 80%-20% theoretical split associated with the 5-year return period.

It should be noted that the results from the training/test data split differed from the ones proposed by Frame et al. (2022). In their study, the frequency analysis was done with instantaneous peak flow observations taken from the USGS NWIS (Survey, 2016), and a maximum cap of 13 water years was used to train each basin. Instead, we used the observed daily data from the CAMELS-US dataset and did not impose restrictions on the maximum number of training years. We would also like to re-emphasize that training the model exclusively on water years containing events smaller than a 5-year return period, and testing it on water years with events larger than 5-year return period, was meant as a form of stress-test to get an intuition of the model behavior regarding extreme streamflow events. In practical applications, one would not choose to use this type of setup, but one should use all available information about this kind of events for model training.

## 2.2 Data-driven, hybrid and conceptual models

The experiments in this study were conducted using three models: a stand-alone LSTM, the HBV model as a stand-alone conceptual model, and a hybrid approach. Both the LSTM and the hybrid model were trained using the Neural Hydrology (NH)

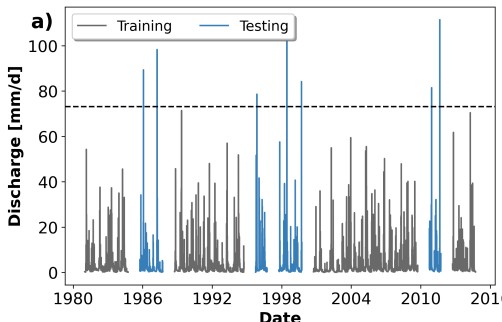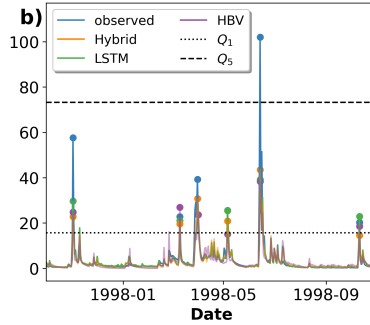

**Figure 1.** a) Observed discharge in mm/day from 1984 to 2016 for catchment id 01054200. Grey lines represent training data, including discharge below the 5-year return period threshold, marked by the dashed line $Q_5$. Blue lines indicate test data for discharge exceeding this threshold. This training/test split, based on discharge exceedance probability, is designed to assess model performance under extreme hydrological conditions. b) Example of peak identification for basin 01054200 in one year of the test period. $Q_1$ represent the one-year return period threshold, which was used to identify peak events. $Q_5$ represent the five-year return period threshold, which was used for the training/testing data split.

package (Kratzert et al., 2022), while the optimization of the stand-alone conceptual model used the SPOTPY library (Houska et al., 2015). Consistent with previous studies, the LSTM and hybrid models were trained regionally, using the information from all 531 basins at the same time, while the stand-alone conceptual model was trained basin-wise (locally). In other words, in this study we compare model results of an LSTM network, a hybrid model, and 531 individually trained conceptual models.

### 2.2.1 LSTM

The hyper-parameters for the stand-alone LSTM were taken from Frame et al. (2022). We used a single-layer LSTM with 128 hidden states, a sequence length of 365 days, a batch size of 256 and a dropout rate of 0.4. The optimization was done using the Adam algorithm (Kingma and Ba, 2014). An initial learning rate of 1e-3 was selected, which was decreased to 5e-4 and 1e-4 after 10 and 20 epochs, respectively. The basin-averaged NSE loss function proposed by Kratzert et al. (2019) was used for the optimization. In a slight deviation from Frame et al. (2022), we trained our model for 20 epochs instead of 30. We trained our models using 5 dynamic inputs from the daymet forcing: precipitation (mm/day), average short-wave radiation (W/m$^2$), maximum temperature (C), minimum temperature (C), vapor pressure (Pa), plus 27 static attributes, listed in Table A1 of Kratzert et al. (2019), describing climatic, topographic, vegetation and soil characteristics of the different basins.

We used an ensemble of 5 LSTM networks to produce the final simulated discharge. In other words, we trained 5 individual LSTM models, with the architecture described above, but initialized each one using a different random seed. After training, we ran each model, individually, to retrieve the simulated discharges, and took the median value as the final discharge signal that we used in the analysis. Using an ensemble of LSTM networks allows us to produce more robust results (Kratzert et al., 2019; Lees et al., 2021; Gauch et al., 2021) and reduces the effects associated with the random initialization of the models.

### 2.2.2 Hybrid model: LSTM+HBV

For the hybrid model architecture, we used the $\delta_n(\gamma^t, \beta^t)$ model proposed by Feng et al. (2022). In this architecture, an ensemble of 16 HBVs acting in parallel was parameterized by a single LSTM network. Each of the 16 ensemble members contained an HBV model with 4 buckets, whose flows were controlled by 11 static plus two time-varying parameters. The discharge of the ensemble was calculated as the mean discharge of the 16 members. Moreover, to produce the final outflow, the ensemble discharge was routed using a two-parameter unit hydrograph. In total, the LSTM produced 210 parameters (16 ensemble members, each with 13 HBV parameters plus two routing parameters) which were used to control the ensemble of conceptual models plus the routing scheme. The model was trained end-to-end. Figure A1 in Appendix A shows a scheme of the hybrid model structure.

During training, each batch contained 256 samples, each with a sequence of 730 days. The first 365 days were used as a warmup period, to stabilize the internal states (buckets) of the HBV and reduce the effect of the initial conditions. These 365 values did not contribute to the loss function. The remaining 365 time steps were used to calculate the loss, backpropagate the gradients, and update the model's weights and biases. Further details on the model implementation can be found in the *hybrid_extrapolation_seed#.yml* files of the supplementary material. To validate our pipeline, we benchmarked our hybrid model implementation using the experiments proposed by Feng et al. (2022). These results are shown in Appendix A, where we show a similar performance between our implementation and their results. Only after validating our pipeline, we ran the extrapolation experiments.

### 2.2.3 Stand-alone conceptual model: HBV

To have a full comparison of the model spectrum, we also included a stand-alone conceptual model. We used a single HBV model plus a unit hydrograph routing routine, resulting in a model with 14 calibration parameters (12 from the HBV plus 2 from the routing). Note that this HBV instance has one fewer parameter (12 instead of 13) than the version used in the hybrid model. This one parameter difference is to maintain consistency with Feng et al. (2022), where the authors used the 13-parameter HBV only when dynamic parameterization was included, and the 12-parameter model for the static version. Similar to Acuña Espinoza et al. (2024), the stand-alone conceptual models were trained basin-wise, using Shuffled Complex Evolution (SCE-UA) (Duan et al., 1994) and Differential Evolution Adaptive Metropolis (DREAM) (Vrugt, 2016), both implemented in the SPOTPY library (Houska et al., 2015). We then selected, for each basin, the calibration parameters that yielded better results.

### 2.3 Performance metrics

To evaluate the overall performance of the model we used Nash–Sutcliffe efficiency (NSE) (Nash and Sutcliffe, 1970). However, the main objective of this study is to evaluate the ability of the models to predict high-flow scenarios. Therefore the majority of the analyses were done using only peak flows.

Given the amount of data comprised in the test period (3 429 years over the 531 basins), the peak identification was done automatically. For this task, we used the *find_peaks* function of the *signal* module in the *SciPy* library (Virtanen et al., 2020), defining a 7-day window as the criterion for independent events. Moreover, we selected only the peaks above the one-year return period threshold, to have a better representation of high-flow scenarios. After we identified the peaks in the observed discharge series, we extracted the associated values from the simulated series of the different models. Figure 1b exemplifies this process for basin 01054200 in one year of the test period where each dot represents an identified peak. Once the peaks were identified, we calculated the absolute percentage error (APE) as a metric for model performance:

$$APE = \frac{|y_{obs} - y_{sim}|}{y_{obs}}, \tag{1}$$

where $y_{obs}$ and $y_{sim}$ are the observed and simulated discharge, respectively.

## 3 Results and Discussion

After training the models, we performed multiple analyses to evaluate their generalization capabilities to extreme events. The results of these analyses are presented in this section. All the results discussed here are for the test period.

### 3.1 Model performance comparison for whole test period

Figure 2 shows the cumulative distribution functions (CDF) for the NSE reported for each model, over the whole test period. We can see that the LSTM outperforms the hybrid model, with a median NSE of 0.75 and 0.71 respectively. Moreover, both models outperform the stand-alone HBV model, which has a median NSE value of 0.64. The result that both the LSTM and the hybrid model outperform the stand-alone HBV is not surprising, and similar results have been reported in the literature (Kratzert et al., 2019; Lees et al., 2021; Loritz et al., 2024; Feng et al., 2022; Acuña Espinoza et al., 2024). This can be attributed to the fact that conceptual models present a relatively simple structure that in a lot of cases oversimplifies the actual physical processes. For example, the HBV model assumes that all flows have a linear relationship with the storage, that the storage/discharge rate does not change over time, and that snow melting is a linear process, proportional to the difference between a threshold temperature and the air temperature. Both the LSTM and hybrid model have more flexible frameworks that allow them to increase their performance. Moreover, we show that even with a different training-test split than the usual temporally-contiguous subsets, our results are consistent with the ones reported by Feng et al. (2022) and Acuña Espinoza et al. (2024), where the same model ranking was observed.

### 3.2 Model performance comparison for peak flows

The metrics shown in Fig. 2 were calculated using the whole test period. Consequently, they summarize the overall performance of the three models. However, the main objective of this study is to evaluate the ability of the models to predict high-flow scenarios. For this, we used the APE metric, which we calculated following the procedure described in section 2.3. Figure 3a presents, for each model, the distribution of the APE for all the peak flows. This figure shows a similar distribution for the three

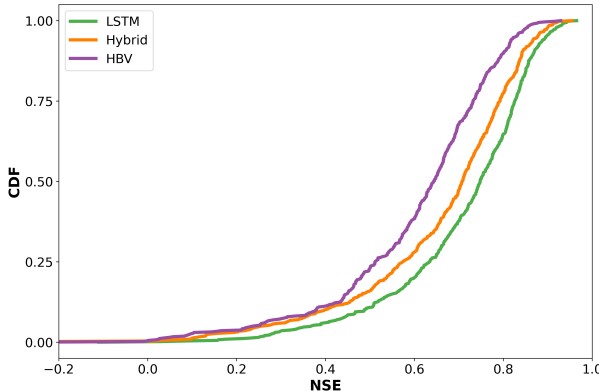

**Figure 2.** Cumulative density function (CDF) of the Nash–Sutcliffe efficiency (NSE) for the different models, generated using 531 basins of the CAMELS-US dataset. The NSE was calculated over the whole test period of each basin.

models, with the LSTM presenting a slightly lower median error than the hybrid and stand-alone HBV. Lower values are better for the APE metric. The finding that LSTMs outperform process-based models aligns with Frame et al. (2022) and helps to challenge the notion that data-driven methods are less capable of extrapolation (Reichstein et al., 2019; Slater et al., 2023). In the case of the hybrid model, although the LSTM exhibits a slightly lower median error, the error distributions of both models are similar. Therefore, we do not find strong evidence suggesting that one architecture is significantly better than the other in this scenario, leaving it to the reader's discretion to choose the model that best suits their needs.

### 3.3 Model performance comparison for out-of-sample peak flows

Figure 3a allowed us to evaluate the performance of the models only in peak discharges. However, this still did not give us a performance metric for values exclusively outside of the training range. More specifically, the results presented in Fig. 3a evaluated the error in 17 580 observed events. Considering that the test period contained 3 429 years, we got an average of 5 peaks per year. However, as shown in Fig. 1b, these peaks were not necessarily larger than the 5-year return period thresholds used during training. Figure 3b shows the same error metric but classifies the peaks based on their return period. The four categories to the right of the dashed vertical line present the errors associated with discharges beyond the 5-year return period threshold, giving a strict evaluation of the generalization capabilities of the models. We can see that the LSTM slightly outperformed the hybrid and HBV models in the 1-5 and 5-25 return period. In the remaining three intervals the performance of the LSTM and hybrid are comparable, with the HBV also showing similar behavior for the last one. Figure B1 in Appendix B shows the variation in the APE due to five random initializations of the models. We can see that in the last three categories, the LSTM performed better in three cases, the hybrid in seven and they both reported the same median value in four. Moreover, the differences in the median value between the LSTM and the hybrid model are smaller than the metric variation due to the random initialization, supporting our hypothesis that for higher return periods all models perform similarly.

In most cases, the errors increased for higher return periods. This was expected as models were trying to generalize to flows farther away from their training range. On the other hand, the 100+ return period peaks presented similar or slightly lower errors than the ones in the 50-100 category. At this point, the reported errors were close to 60%, which indicated that no model could satisfactorily reproduce the observed peaks. Moreover, because of the characteristic of the metric (see Eq. 1), the error was scaled by the magnitude of the observation, which would explain why the 50-100 and 100+ presented similar errors.

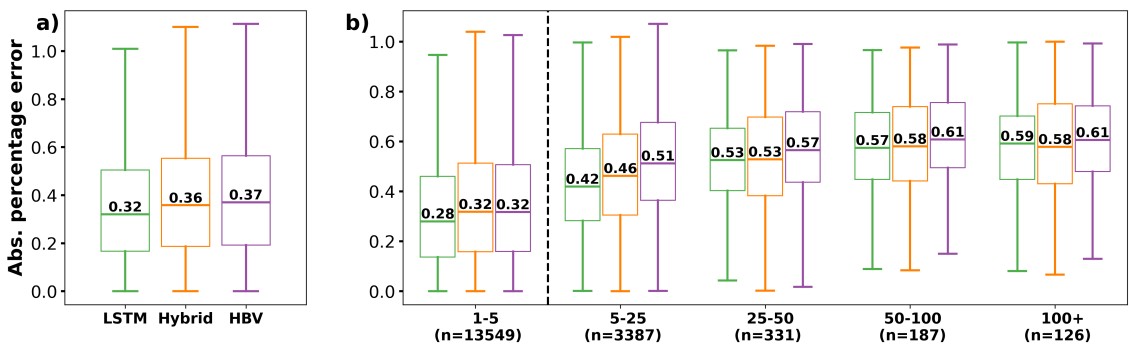

**Figure 3.** a) Absolute percentage error (APE) between the observed peak discharge and the associated simulation value for the different models. The results show the error distribution, from all 531 basins, calculated only for the peak flows of their test period (total of 17580 values) b) APE, classified by the return period of the observed peaks. The four categories to the right of the dashed vertical line present the errors associated with observed discharge above the 5-year return period threshold, evaluating the out-of-sample capabilities of the models. The n-value below each category indicates the amount of data used to produce the box-plot.

### 3.4 Spatial analysis of model performance

To further understand the capacity of the different models, we evaluated their performance in space, examining how predictive accuracy varies across regions and identifying differences between the models. As shown in Appendix C, in Fig. C1, all models exhibited lower performance in the basins located in the Great Plains (center) and southwest of the US. This pattern, previously documented in the scientific literature for both data-driven methods (Gauch et al., 2021) and process-based models (Newman et al., 2015), is also observed in the hybrid architecture, as demonstrated here. Martinez and Gupta (2010) identified aridity ($PET/P$) and runoff ratio ($Q/P$) as good predictors of model performance. The comparison between Fig. C1 and Fig. C3 supports this hypothesis, revealing an association between lower performance and regions characterized by high aridity and low runoff ratios.

Given the shared behavior between the models, and in line with the objective of this comparison study, we further evaluate the differences in their performance. Figure 4 shows the difference between the hybrid and the LSTM model, calculated as $APE_{\text{Hybrid}} - APE_{\text{LSTM}}$. Consequently, negative (blue) values indicate that the hybrid model performed better (i.e., presented lower error), while positive (red) values favor the LSTM.

The first spatial tendency that we can notice is that, along the central part of the US, the LSTM tends to perform better. As previously discussed, this area is characterized by arid basins. In arid basins, surface runoff can be associated with high-intensity events over a short time period. Moreover, the HBV model generates runoff under the assumption that the discharge is a function of the basin storage, and lacks a direct channel to transform precipitation into surface runoff (e.g., the water always routes through a linear reservoir). This assumption might not hold for the runoff-generating process on arid basins. Considering that the hybrid model is regularized by an HBV layer, this structural deficiency would explain why the LSTM outperforms the hybrid model in this area.

The second spatial tendency that we can observe is that, for the 100+ return period map, there is a cluster on the Pacific Northwest in which the hybrid model outperforms the LSTM. These basins are characterized by high precipitation values (see Fig. C3c) and high discharges. This is a challenge for the LSTM architecture, due to saturation problems, which will be explained in detail in the next section.

Figure C1 further presents the differences between the HBV and LSTM models as well as between the hybrid model and HBV. In the first case, the LSTM outperforms the HBV in most cases, with the same exception of the northwest coast for the 100+ return period. In the second case, the hybrid model presents an overall better performance.

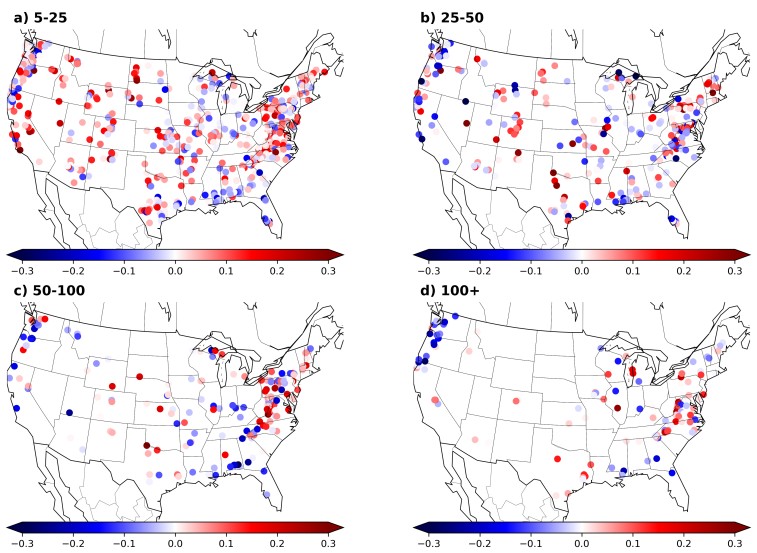

**Figure 4.** Spatial visualization of the difference between the absolute percentage error (APE) for the hybrid and LSTM models. Each point corresponds to one basin. The four maps are associated with the different return periods. The color scale indicates the difference between the median APE for the different models. The difference is calculated as $APE_{\text{Hybrid}} - APE_{\text{LSTM}}$, therefore negative (blue) values indicate that the hybrid model performs better and positive (red) values indicate that the LSTM performs better.

## 3.5 Saturation analysis: behavior of the models during extreme flow scenarios

The saturation problem in LSTM models with a single linear head layer (as described by Kratzert et al., 2024) arises due to the inherent limitations of the model architecture, resulting in a theoretical prediction limit (see equation B2 of Kratzert et al., 2024). In other words, independently of the input series, the associated prediction cannot go above the theoretical limit. This limit is a function of the weights and biases of the head linear layer and varies for each trained model instance. From our experiments, where the model is trained only on discharge values below a 5-year return period, we observed that the maximum value predicted by the LSTM was 78.9 mm/day, which is close to the calculated theoretical prediction limit of 83.9 mm/day. This saturation limit could explain the LSTM's low performance for the cluster of basins in the Pacific Northwest of the US, for the 100+ return period case. Out of the ten basins with the highest errors within this cluster, seven of them presented 100+ return period discharges above the saturation limit of the LSTM. Therefore, independently of the forcing series, the model was not able to reach such values. Figure 5a shows an example in which this saturation problem is particularly pronounced. On the other hand, both the hybrid and the stand-alone HBV model do not have such a theoretical limit. The conceptual model architecture is defined with an unlimited capacity in the buckets, and due to mass conservation, the water received by the models, after evapotranspiration and other abstractions, must be accounted for within the system.

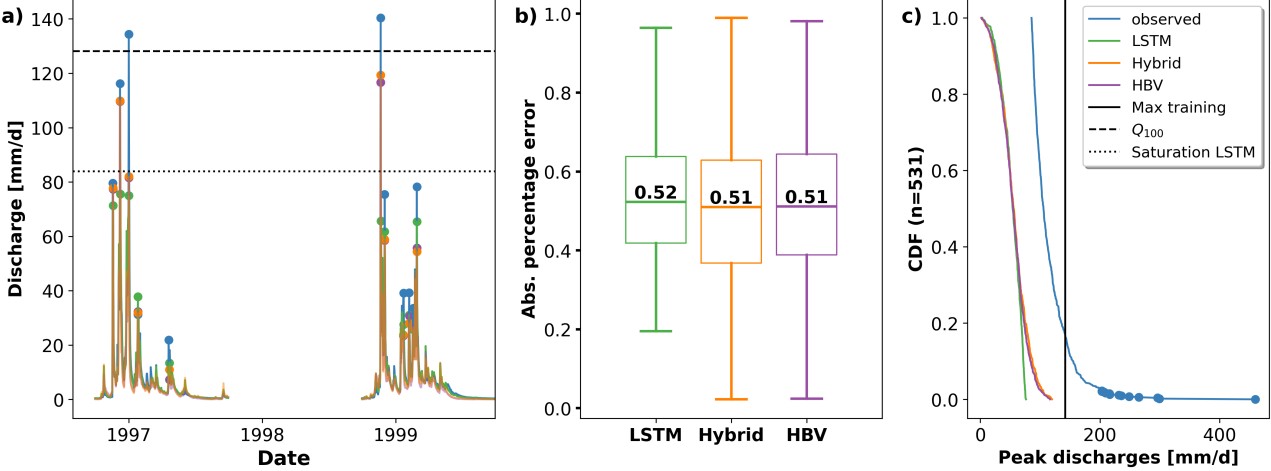

**Figure 5.** a) Observed and simulated discharges, for two years of the testing period, for basin 11532500. The dashed and dotted lines indicate the 100-year return period discharge and the theoretical saturation limit of the LSTM model, respectively. b) Absolute percentage error (APE) of the 531 highest discharges for the different models. c) Cumulative density function (CDF) of the 531 observed highest discharge values across all basins and their respective simulated values. The blue dots help visualize that less than 3% of the events have values between 200 and 400 mm/day.

The theoretical prediction limit in the LSTM is a function of the weights and biases of the head layer, which are a result of the training process. In our experiment we artificially restricted the training data to discharges smaller than the 5-year return period thresholds, reducing the support of the data space the model was fitted to. Consequently, this setup directly intensified

the saturation problem. In practical applications where the model is trained on all available data, the saturation issue would tend to decrease in relevance. Moreover, it would only affect the few gauges in which the extreme discharges are above the saturation limit. However, a theoretical saturation limit remains, which is an undesirable property in a hydrological model, especially in cases where we are designing infrastructure for extreme events outside of any training data (e.g., 1 000-year flood). Further research should be invested in overcoming this problem.

Apart from the statistical artifacts introduced by our selection procedure, we found two potential issues that might lead to the peak underestimation of the hybrid models. Figure 6 shows the precipitation and observed discharge, together with the accumulated value and the simulated discharge series, for 4 of the largest events in the dataset.

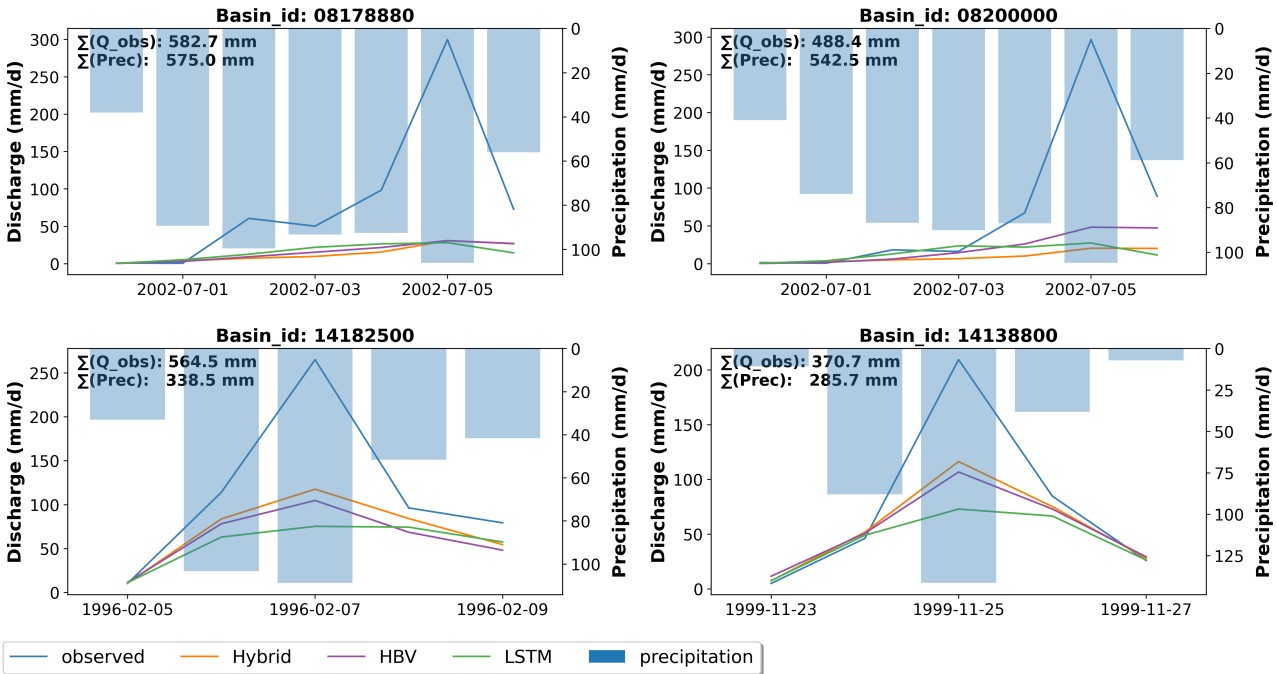

**Figure 6.** Example of 4 of the most extreme events presented in the dataset. The subplots show the precipitation series and the observed and simulated hydrographs. $\sum$Q_obs and $\sum$Prec indicate the cumulative sum of the discharge and precipitation series. Basins 08178880 and 08200000 have similar precipitation and discharge volumes while basins 14182500 and 14138800 have a precipitation volume smaller than the discharge volume.

For basins 08178880 and 08200000, the accumulated precipitation of the event is similar to or larger than the accumulated discharge, however, the simulated series strongly underestimates the discharge. This behavior can arise due to structural limita-
tions in the hydrological model. For example, given the lack of a fast response channel, a high precipitation pulse can be divided and routed through several linear reservoirs, attenuating the respective discharge peak. This effect could have been strengthened by our training/test split, given that the optimization parameters, which control the interaction between the buckets, were learned for certain conditions, which were inadequate for other out-of-sample hydrological events.

In this regard, the hybrid model presents a theoretical advantage over the HBV model through the possibility of dynamic parameterization that adapts the model behavior to current conditions. The $\delta_n(\gamma^t, \beta^t)$ hybrid model uses a dynamic $\beta$ coefficient to control the recharge rate at which precipitation was transferred to the other buckets. However, as discussed in the previous paragraph and as shown in Figure D1, during high-intensity events $\beta$ reached the limits of its predefined interval, which limited the model to further adapt its behavior.

The second issue that we found is a possible bias in the input data. For basins 14182500 and 14138800, the accumulated precipitation is smaller than the accumulated discharge, which would explain the underestimation of the simulated values. Westerberg and McMillan (2015) indicate that bias in precipitation measurements can be caused by point uncertainty, interpolation uncertainty, and equipment malfunction. Moreover Bárdossy and Anwar (2023) indicate that catchment-averaged precipitation values present higher bias during extreme events. This bias poses a challenge for the HBV and hybrid model, which relies on a mass-conservative structure. Without sufficient water input to the system, the model inherently cannot replicate the observed discharges.

An alternative hypothesis to precipitation bias is that the high discharges are caused by snow or glacier melt. In this case, the accumulated precipitation during the event can be smaller than the accumulated discharge, since part of the discharge would be caused by melting water that entered the system weeks or months before. Both basins are located in the state of Oregon (north-west of the USA), and, accounting for the dates of both events, there is a possibility for snowmelt-induced discharge. Nevertheless, the snow module of HBV is not reproducing this behavior, which would point towards a structural deficiency in the model.

### 3.6 Limitations and uncertainties

In this study, we focus our results and analysis on model performance, as we tackled our research questions from a practitioner's point of view. However, it should be noted that other criteria, such as model interpretability, also play an important role in an integral model evaluation, as it allows us to assess whether the generated results align with expected domain-specific behaviors. Appendix D briefly discusses the temporal variation of the dynamic parameters for the hybrid model in this context. We further refer to studies such as Acuña Espinoza et al. (2024), Kraft et al. (2022), Höge et al. (2022) and Feng et al. (2022) for a deeper analysis of model interpretability for hybrid models.

Following the procedure proposed by Frame et al. (2022) we split the training and test periods by years, based on the return period of the maximum annual discharge. Both periods used information from the previously selected 531 basins. In future work one could use different subsets of basins during training and testing, to further compare the models on an ungauged basin scenario and evaluate their spatial generalization capability.

The comparison presented here was done using the hybrid architecture proposed by Feng et al. (2022). This architecture was chosen because it gave a competitive performance with LSTMs in their original experiment and because the code was open source. However, other architectures might lead to different results. From the analysis presented above, we hypothesized that a process-based layer that includes a fast-flow channel might improve performance. Moreover, expanding the hybrid model's parameter range, or the number of dynamic parameters, might be beneficial. However, the added flexibility might come at the

expense of model interpretability. Additionally, considering that catchment-averaged precipitation values present higher bias during extreme events, strategies to overcome this limitation should be tested. Given the non-mass-conservative structure of the LSTM, systematic input biases can be accounted for. However, similar strategies should be evaluated for the hybrid case (e.g. considering a dynamic parameter that factorizes the precipitation input). Furthermore, other strategies to create a hybrid model, such as component replacement, should also be tested (Kraft et al., 2022; Höge et al., 2022). We encourage the hydrological community to expand the test cases presented here.

The training/test split applied in this study was intended as a form of stress-testing, to get an intuition of a model's capacity to generalize to unseen events. For the reasons stated in previous sections, this stress-testing method directly affects the saturation problem in the LSTM and the parameter optimization for the hybrid and conceptual models. In a practical case, one should not remove low-probability events from the training data. Furthermore, in an operational setting, all the data should be included during model training, to increase the performance of the models (Nevo et al., 2022).

Lastly, differences between simulated and observed values, especially in extreme events, can also be attributed to higher uncertainty in the observed quantities, including discharge and precipitation (Di Baldassarre and Montanari, 2009; Westerberg and McMillan, 2015; Bárdossy and Anwar, 2023). We did not consider this type of uncertainty in our analysis, as this would be outside of the scope of the paper.

## 4    Summary and conclusions

In this study, we evaluated the generalization capabilities of data-driven, hybrid, and conceptual models for predicting extreme hydrological events. Following the methodology proposed by Frame et al. (2022), we partitioned our data based on the occurrence frequencies, using the 5-year return period discharge as a threshold. We trained our models using information from water years with discharges strictly lower than the threshold and tested their performance on low-probability data. This setting was meant as a form of stress-test to get an intuition of the model behavior regarding extreme streamflow events. Our findings indicated that the LSTM slightly outperforms the hybrid and HBV models for 1-5 and 5-25 return periods, and all models show similar performance for higher discharges.

The spatial analysis of the models' performance revealed that all three models exhibited higher errors in more arid basins, consistent with findings reported in the literature (Martinez and Gupta, 2010; Gauch et al., 2021; Newman et al., 2022). Regarding the differences between the models, the LSTM presented lower errors than the hybrid model in more arid basins, particularly for events in the 5-25 and 25-50 return period categories. This disparity can be attributed to the structural limitations of the HBV model, which assumes that discharge is a function of basin storage, a premise that may not align with runoff-generating processes in arid basins. Given that the hybrid model is regularized by an HBV layer, this structural deficiency would explain why the LSTM outperforms the hybrid model in this area. On the contrary, for the 100+ return period events, the hybrid model outperformed the LSTM in a cluster of basins located on the northwest coast of the US. This behavior can be linked to the LSTM's theoretical discharge saturation limit, which is determined during training. In the northwest cluster, the event's magnitude exceeded the saturation limit of the LSTM, preventing the LSTM from simulating the observed discharges.

As discussed before, the training/test split used in this study artificially increased the saturation problem, as it constricted the data space the model was fitted to. In practice, this problem can be attenuated by also considering low-probability events during training. Nevertheless, a theoretical limit will still exist, which is not a desirable property for a hydrological model. Additional research to overcome this limitation should be encouraged.

Expanding on the analysis to include the most extreme scenarios across the whole dataset, it can be concluded that all models underestimated the most extreme flow scenarios. However, the hybrid model and HBV were able to simulate higher discharges than the LSTM, and presented an error distribution with longer tails toward smaller values. Upon further investigation, we noticed that the reasons for underestimating the extreme flow scenarios were different. For the LSTM, its saturation limit was reached. On the other hand, the hybrid and HBV models underestimated the discharge due to structural deficiencies and

possible bias in the input data. The dynamic parameterization of hybrid models might help reduce the former, by changing the model response based on current conditions. This idea is conceptually similar to how an LSTM operates, in which the gate structures operate based on current and past conditions.

Overall, in most of the experiments performed here, we did not find strong evidence suggesting that there is a significant difference between the extrapolation capabilities of LSTM networks and hybrid models. However, hybrid models did report

slightly lower errors in the most extreme cases and were able to produce higher peak discharges. We leave it to the reader's discretion to choose the model that best suits their needs.

## Appendix A:  Benchmarking Hybrid model

For our Hybrid model we used the $\delta_n(\gamma^t, \beta^t)$ architecture proposed by Feng et al. (2022). A scheme of this architecture is shown in Fig. A1. Given that our experiment pipeline was executed in the Neural Hydrology package, we first had to benchmark our model implementation against the original case. Figure A2 shows that our model implementation produced similar results to the one reported by Feng et al. (2022).

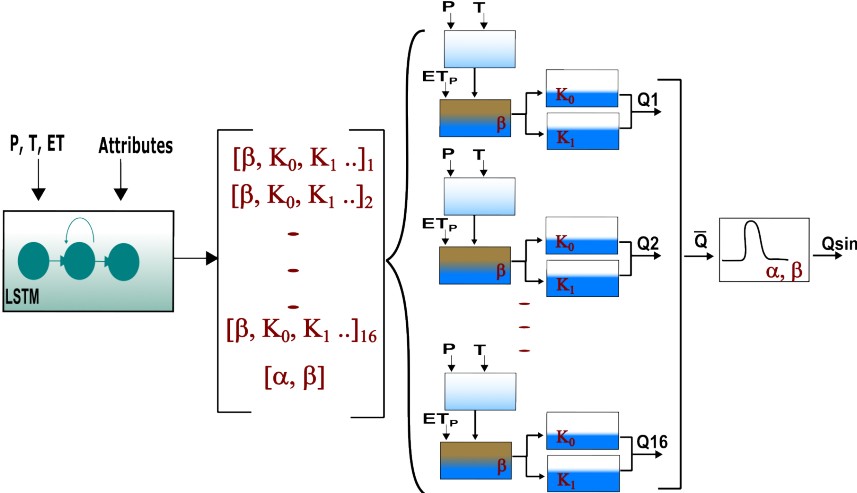

**Figure A1.** Scheme of the hybrid model structure. The LSTM predicts the parameters used to parameterize the different hydrological modules. In total, the LSTM produced 210 parameters (16 HBV members each with 13 parameters, plus two routing parameters). The 16 hydrological models are run in parallel. The produced discharges are averaged, and the averaged signal is further routed using a unit hydrograph based on the gamma function. The output of the routing module is retrieved as the final simulated discharge.

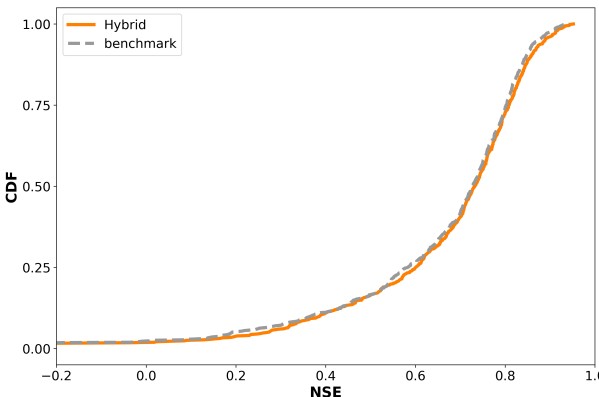

**Figure A2.** Cumulative density function (CDF) of the Nash–Sutcliffe efficiency (NSE) for different models, generated using 671 basins of the CAMELS-US dataset.

## Appendix B: Impact of random model initialization on absolute percentage error metric

Figure B1 shows the effect of different model initializations, in the APE metric. The ranking of the models in the last three categories (25-50, 50-100, 100+) varies depending on the model initialization. This indicates that the differences in the median values are within the statistical noise, and we cannot conclude that one model is better than the other.

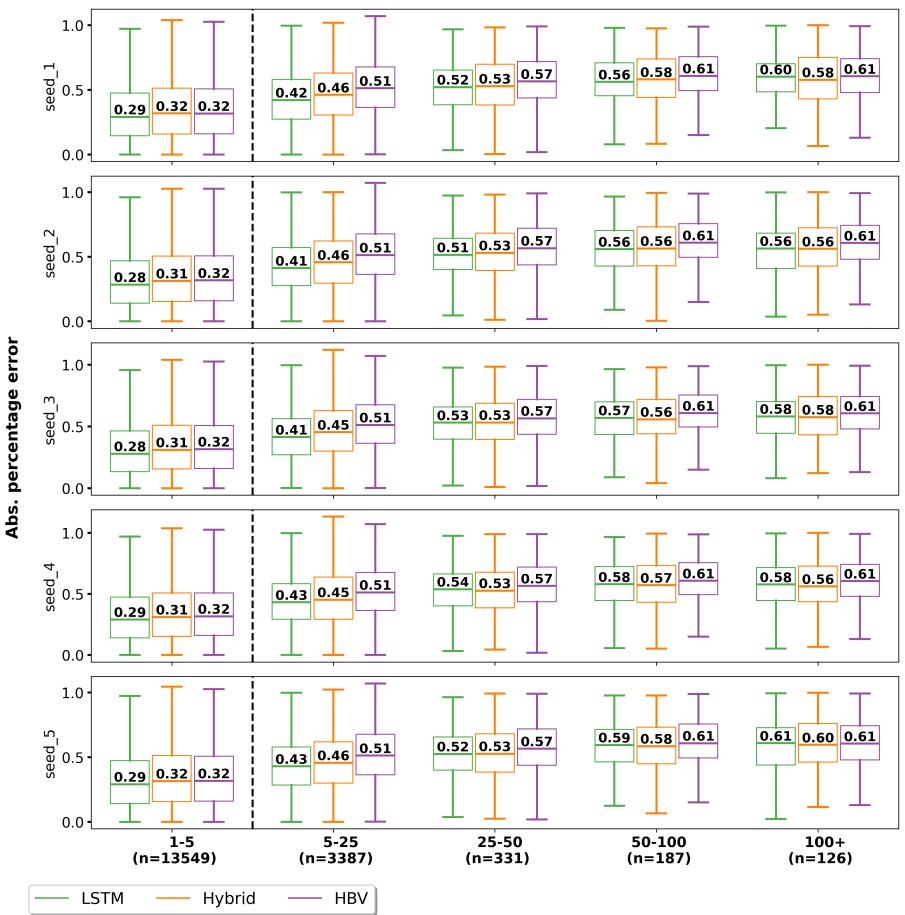

**Figure B1.** Variation in absolute percentage error (APE) due to random initialization of the LSTM and hybrid models.

## Appendix C: Spatial visualization and comparison of model performance

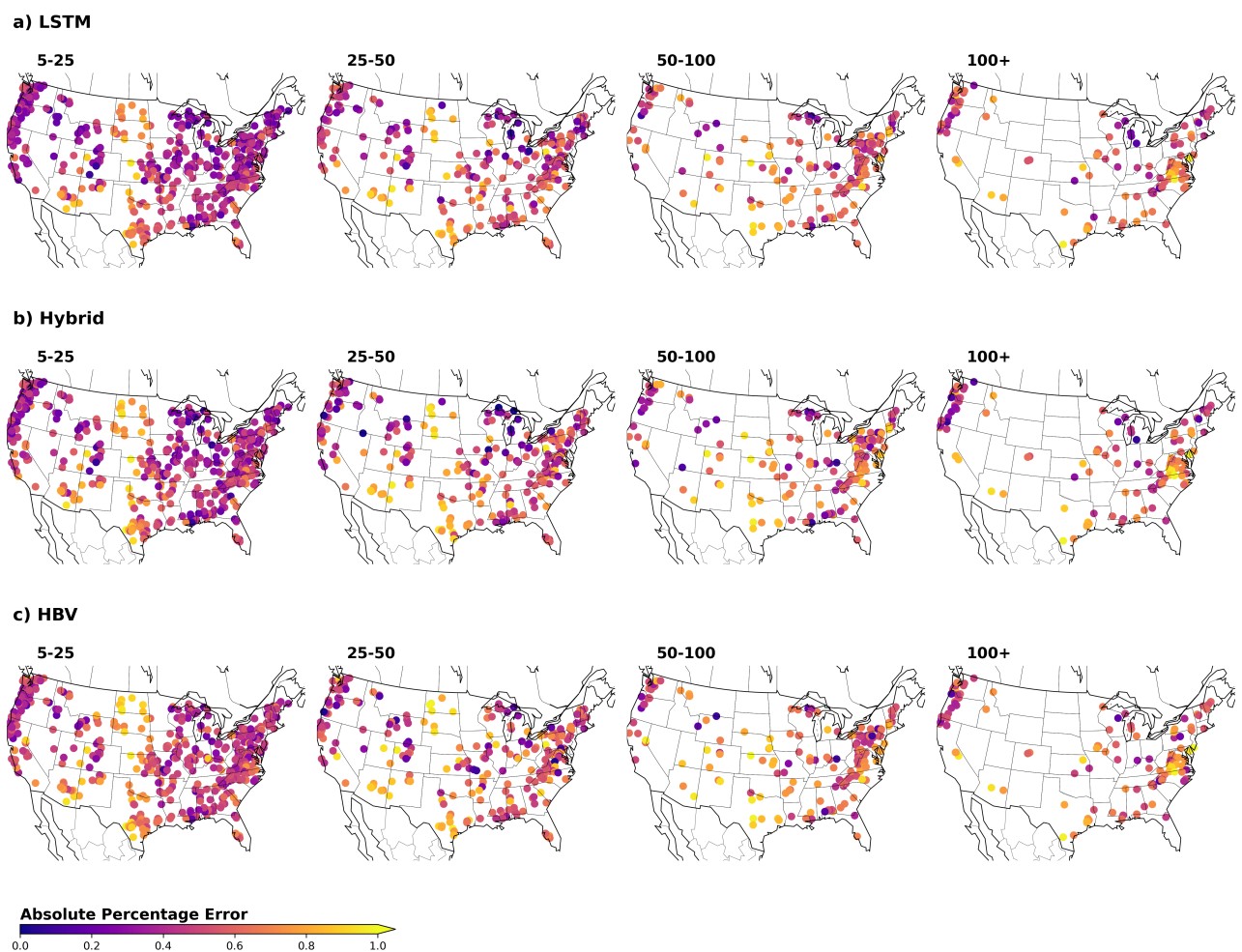

**Figure C1.** Spatial visualization of absolute percentage error (APE), for the different models and the different return periods. Each point is associated with one basin. The scale indicates the median APE between observed and simulated values, for all the events associated with the respective basin and return period.

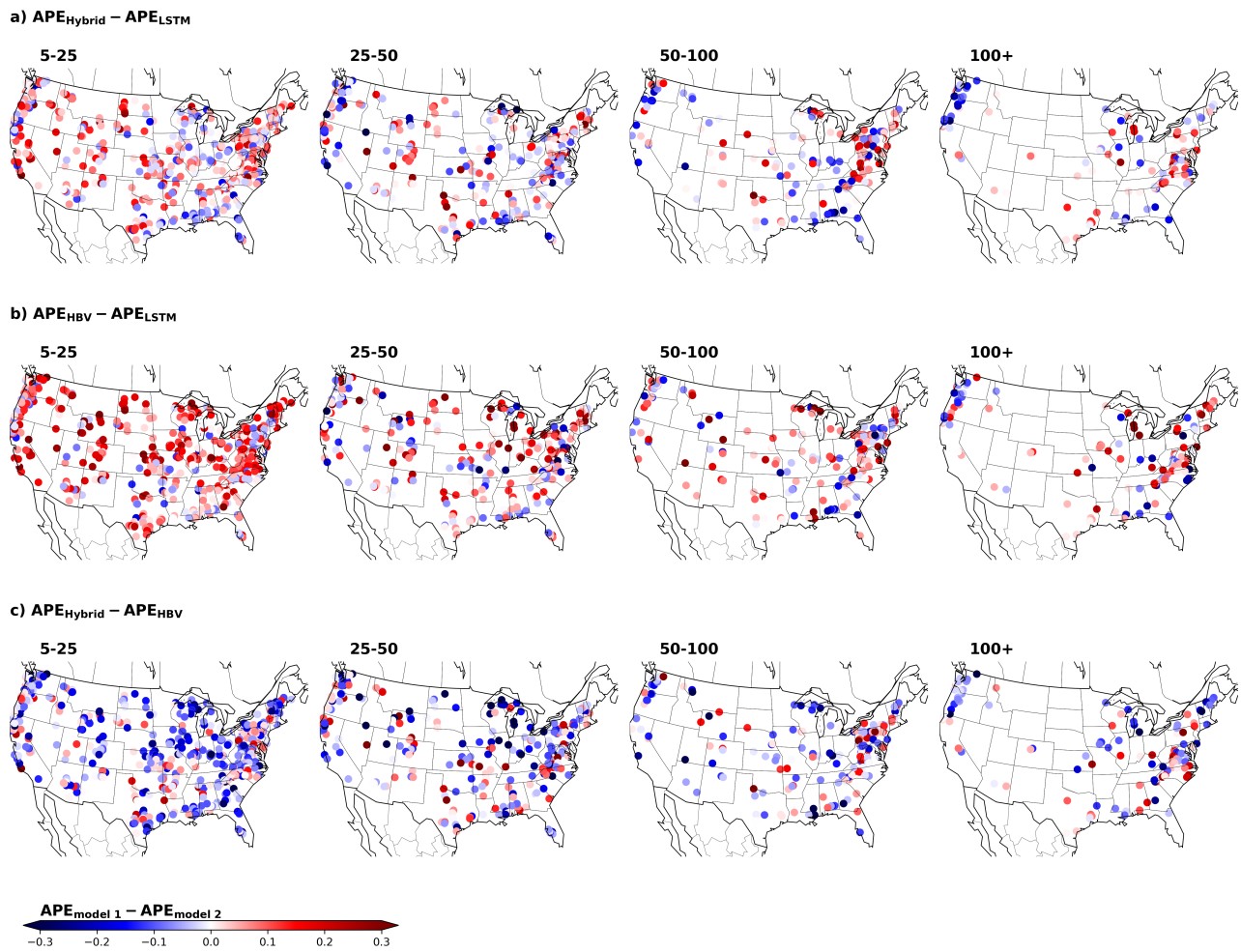

**Figure C2.** Spatial visualization of the difference between the absolute percentage error (APE) for the different models. Each point is associated with one basin. The scale indicates the difference between the median APE of the different models. The difference is calculated in the order in which the models are named, $APE_{\text{model 1}} - APE_{\text{model 2}}$, therefore negative (blue) values indicate that model 1 performed better while positive (red) values indicate that model 2 performed better.

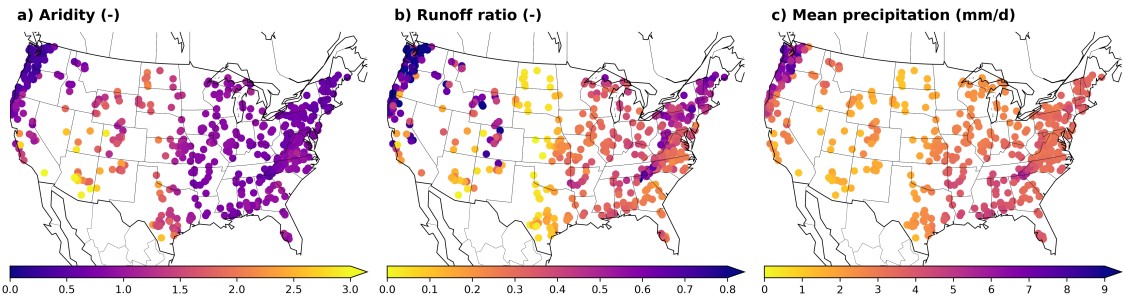

**Figure C3.** Spatial visualization of basin-averaged static attributes. Each point is associated with one basin. a) Spatial variability of aridity b) Spatial variability of runoff ratio c) Spatial variability of mean daily precipitation.

## Appendix D: Temporal variation of dynamic parameters

Figure D1 shows the temporal variation of the dynamic parameters of the hybrid model for three basins. For the first two basins, we can see clear cyclic patterns, in which the parameters adjust in dry/wet seasons to produce less/more water. These patterns were discussed in Acuña Espinoza et al. (2024) for experiments conducted in CAMELS-GB, and suggest the possibility that the LSTM control the HBV models in a consistent way. However, further investigation is needed to understand the LSTM-HBV interaction. In the third basin, the hydrograph does not present such a clear distinction between dry and wet seasons, and we can observe more variation in the parameters.

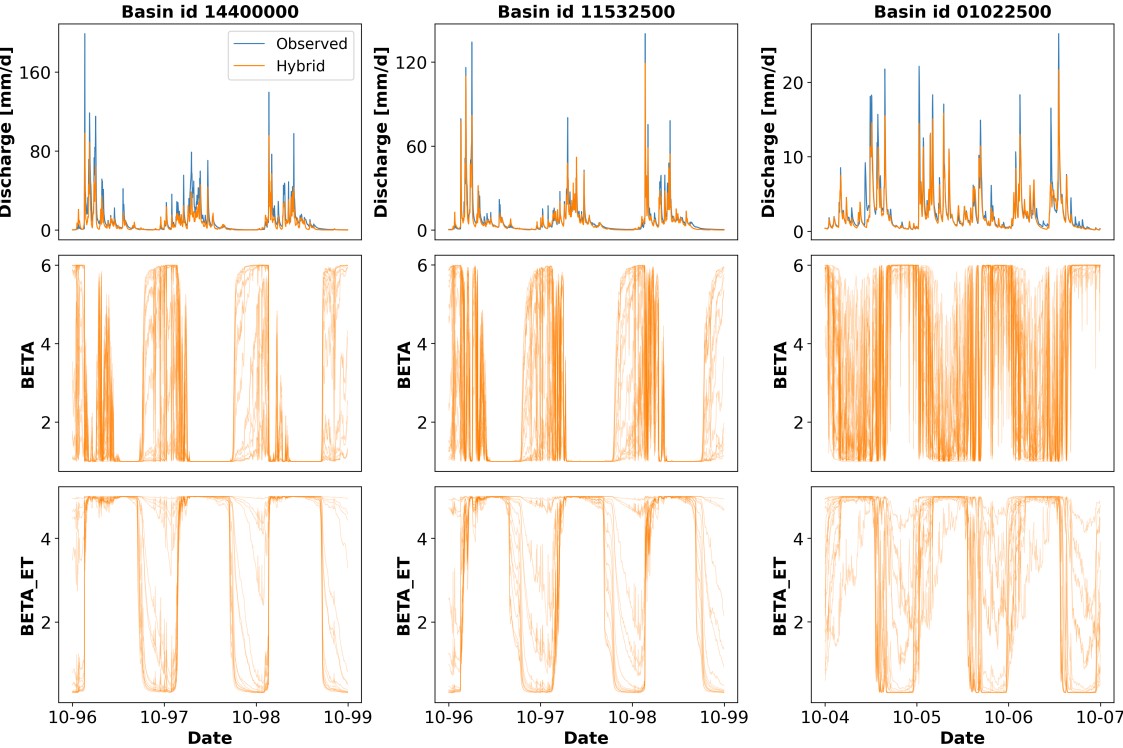

**Figure D1.** Time series indicating the variation of parameters for three different basins and their association with the simulated hydrographs.

*Code availability.* The code used to conduct all the analyses can be found at https://doi.org/10.5281/zenodo.14191623 (Acuna Espinoza, 2024) and as part of our GitHub repository https://github.com/eduardoAcunaEspinoza/hybrid_extrapolation/tree/v0.2

*Data availability.* The CAMELS US dataset is freely available at https://doi.org/10.5065/D6MW2F4D (Newman et al., 2022). All the data generated for this publication can be found at can be found on https://doi.org/10.5281/zenodo.14191623 (Acuna Espinoza, 2024)

*Author contributions.* The original idea of the manuscript was developed by all authors. The codes were written by E.A.E with support from F.K., M.G., and D.K. The simulations were conducted by E.A.E. Results were further discussed by all authors. The draft of the manuscript was prepared by E.A.E. Reviewing and editing was provided by all authors. Funding was acquired by U.E. All authors have read and agreed to the current version of the manuscript.

*Competing interests.* Some authors are members of the editorial board of HESS.

*Acknowledgements.* We would like to thank the referees, Dr. Basil Kraft and Dr. Shijie Jiang, as their input in the review process allowed us to produce a better manuscript. We would also like to thank the Google Cloud Program (GCP) team, for awarding us credits to support our research and run the models. UE would like to thank the people of Baden-Württemberg who, through their taxes, provide the basis for this research.

*Financial support.* This project has received funding from the KIT Center for Mathematics in Sciences, Engineering and Economics under the seed funding program. Daniel Klotz acknowledges funding from the Helmholtz Initiative and Networking Fund (Young Investigator Group COMPOUNDX, grant agreement no. VH-NG-1537).

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
