# Peer review of "Analyzing the generalization capabilities of a hybrid hydrological model for extrapolation to extreme events"

_EGUsphere, 2024_

## Referee Comment (RC1)

**"Analyzing the generalization capabilities of hybrid hydrological models for extrapolation to extreme events"**

**Manuscript #*2024-2147**

September 5, 2024

**Short summary and highlights**

This study examines how well an established hybrid hydrological model generalizes under extreme conditions, comparing its performance to a data-driven model (LSTM) and a conceptual hydrological model. They split the discharge time series into training and test sets based on the magnitude of events within a hydrological year. Normal years were used for training, while extreme ones were used for testing.

The results indicate that the LSTM model performs best across all test data. For extreme events, the performance gap between the LSTM and hybrid models narrows, and eventually disappears for events with a 50-year return period or above. All models face challenges with extremely large discharge events, and the LSTM shows limitations due to saturation, where it cannot produce outputs beyond a certain threshold. While the hybrid and conceptual models slightly better reproduce extreme event patterns, they all exhibit substantial errors under these conditions.

Overall, the study suggests that while hybrid models may offer a modest advantage for modeling extreme discharge events, the LSTM performs better across all conditions. Thus, the choice of the best model should be guided by the specific objectives, such as whether the focus is on overall discharge or extreme conditions, as well as considerations for control and interpretability.

The study offers a critical perspective on hybrid models, which have gained popularity in recent years. While the benefits of hybrid models are often promoted, a thorough evaluation is still needed. Therefore, I find the study highly relevant. The study design is clear, and the use of established models lends credibility to the findings.

**Major remarks**

1. Could you explain why a fixed number of epochs was chosen for training instead of early stopping?

2. The study evaluates a specific hybrid model. While this is valuable, it is important to note that these findings may not apply to all hybrid models. I suggest updating the title to "Analyzing the generalization capabilities of **a** hybrid hydrological model for extrapolation to extreme events", and to discuss this limitation more in-depth.

3. If feasible, a comparison regarding low flow conditions would be appreciated.

4. The overall structure could be improved. Currently, the Results section includes some Method descriptions, and within the Results and Discussion section, results and discussion are not clearly separated into paragraphs.

5. Although it may be beyond the scope of this study, examining the robustness of the (interpretable) hybrid model parameters would be very interesting.

6. The discussion falls a bit short in general. I would appreciate a more in-depth discussion of the findings.

   a) A comparison with existing studies would be beneficial. How do your findings relate to other research in this area, for example the study mentioned by another reviewer, Song at al. (2024), 10.22541/es-soar.172304428.82707157/v1?

b) There are other advantages of hybrid (and conceptual) models over neural networks, such as interpretability. I would appreciate a brief discussion of this aspect.

c) Could these findings apply to other hybrid models and domains, why (not)?

**Minor remarks**

Here I list some typos and suggestions for improving clarity:

L11 Clarify "the latter study".

L9-L13 Consider revising the first paragraph for improved clarity.

L35/L37 Choose either "large events" or "extreme events" for consistency.

L36 Perhaps rephrase to "How does **a** hybrid model compare to **a** process-based model"? See Major remarks.

L38 Specify the type of advantage being discussed.

L41 Rephrase to: "In Section 3, we compare the results of various tests that assess generalization capabilities."

L50 A different name for the hybrid model might be clearer; "$\delta(\gamma^t, \beta^t)$" is somewhat cumbersome.

L53 Consider using "Experimental setup" as the section title.

L60 Move "(a water year is defined as the period from October 1 to September 30)" to where you first mention "water years" on line 59.

L71 Provide a reference for USGS NWIS.

L74ff Please rephrase for clarity.

L89 Add a comma before "respectively".

L89 Define NSE and provide a reference.

L90 Specify which study is referred to as "the original study".

L91 Provide full names and add a space between variable names and units.

L92 Include categories of static variables, such as "27 static variables describing topography, soil properties, and land surface cover ...".

L96 Briefly mention the benefits of using ensemble methods.

L98 Replace "As mentioned in the introduction," with "For the hybrid model architecture,"

L103 Rephrase to: "210 parameters (16 ensemble members, each with 13 HBV parameters plus 2 routing parameters)".

L105 Did you use the same warm-up period for the LSTM, considering it also needs to initialize its states?

L107 Change "365 elements" to "365 time steps".

L108 Provide the full name of the YAML file.

L113 Rephrase: "To ensure a comprehensive comparison of the model spectrum,"

L114 Rephrase to "14 parameters (12 HBV plus 2 routing)".

L115 Note that this HBV instance has 12 parameters, while the hybrid model has 13.

L122 Rephrase to: "their generalization capabilities in the time domain to extreme events."

L124 Consider "Model comparison for the whole test period" as a section title.

L125 Introduce NSE before line 89.

L125 Change "reported **for** each model".

L126 Rephrase: "The LSTM outperforms the hybrid model, with a median NSE of 0.75 and 0.71, respectively."

L127 Rephrase: "The hybrid model has a median NSE of 0.64. This indicates that even with a different training-test split than the usual temporally contiguous subsets, our results align with those reported by Feng et al. (2022) and Acuña Espinoza et al. (2024), where the same model ranking was observed."

Fig. 2 Caption: "Cumulative Density Functions (CDF)".

L131 Consider "Model Comparison for peak flows" as a section title.

L135ff This section sounds more like methods; consider restructuring.

L148 Omit: "This trend will also be observed in other experiments discussed in the following sections."

L150 The sentence is quite generic. Consider removing it and discussing the point in detail later.

Fig. 3 Add ')': "The results of subplot b) show the error distribution."

L168 Use "Eq. 1" instead of "equation 1".

L176 Replace "has to go out". Maybe "has to leave the system"? Also, the water could just stay in the system and accumulate over time, right?

L181 Omit: "This phenomenon as such is not necessarily an indication of model deficiencies."

L228 Replace "For the reasons stated in previous sections, . . ." with "This stress-testing . . ."

L229 "In a practical case, one should use all the data during model training, to increase the performance of the models." Could you mention that you mean using also extreme events, and not literally all data (because we want training/test split)?

L259 Replace "As explained in the manuscript," with "For our hybrid model,"

L260 Please rephrase: "Because our experiment pipeline was executed in the NeuralHydrology package, we **did** first  benchmark our model . . .".

---

## Community Comment (CC1)

The authors (referred to as Espinoza24 thereafter) have made a valuable contribution with their analysis. This comparison provides useful insights. Espinoza24 demonstrates that the LSTM model performed moderately better than the hybrid model (NH-hybrid) for return periods of 5-10, 10-25, and 50-100 years, while the NH-hybrid was slightly superior for return periods exceeding 100 years. They also showed their Hybrid model's results are comparable to observed soil moisture which is encouraging.

We have run similar experiments on our end, which show that our version of single hybrid model, dHBV, outperformed LSTM in nearly all return-period categories. These results are documented here: https://t.co/BnWtEy6NEk. The conclusions seemed to be modestly different from Espinoza24.

To understand where discrepancies lie, we performed extensive due diligence by running multiple experiments with the same setups as the authors to understand the observed differences. We appreciate the authors for making their code available, enabling this exploration. Our experiments used the same data split as Espinoza24. The models compared include:

● **LSTM**: NeuralHydrology version of LSTM from Espinoza24.
● **NH-Hybrid**: Differentiable HBV from Espinoza24 based on the NeuralHydrology package.
● **dHBV1.0 hydroDL**: Differentiable HBV by Feng et al., 2022, based on the HydroDL package.
● **dHBV1.1p hydroDL**: An improved version with updates to the loss function and capillary terms from dHBV1.0 hydroDL.
● **HBV**: Traditional HBV.

A notebook reproducing the results of dHBV1.0 hydroDL (blue box in Figures C1 and C2) is available here (https://colab.research.google.com/drive/12xUvTu9NoGVdcRWqJvypy4DGg5jy5q9T#scrollTo=v0JIEuFZjkxq).

Both dHBV1.0 hydroDL and dHBV1.1p hydroDL showed advantages over LSTM and NH-Hybrid, particularly for high peaks (Figure C1) and the 531 largest peaks (Figure C2), with the benefits being more pronounced for larger return-period event. Notably, NH-Hybrid exhibited larger peak errors than LSTM for the 25-50 and 50-100 return periods, while the HydroDL versions showed smaller errors. This discrepancy may influence perceptions of the relative strengths of these models.

Further, we observed that alternative choices (Table C1) in experimental design, which might align more closely with Frame et al., 2022, or represent more realistic tests, could yield greater advantages for dHBV HydroDL over LSTM (Figure C3). Specifically, the experiments in Espinoza24 may allow LSTM to operate more within an "interpolation" regime, whereas true experiments should challenge models in an "extrapolation" context.

Additionally, Espinoza24 interpreted the 16 multicomponents in dHBV as an ensemble, but this setup is more analogous to the "hidden size" concept in LSTM. Due to the process-based

nature of differentiable models, a true ensemble should consist of different model structures (e.g., HBV, SAC-SMA, PRMS, CFE). Preliminary results (not shown here, but to be provided in a subsequent publication) indicate that such structural variations improved NSE metrics.

While Espinoza24 conducted an experiment to verify that NH-hybrid could reproduce earlier results from Feng et al. (2022), it is important to note that this does not imply that other experiments would yield the same outcome. The claimed equivalence, which could influence how some readers interpret the results, is not established here.

In conclusion, while the LSTM model in Espinoza24 represents a substantial effort, it may not reflect the state-of-the-art due to differences in training frameworks and frontend LSTM configurations. The community may benefit from more explicit specification of implementations used, pulling the original dHBV1.0 HydroDL code into the comparison, and that alternative methodologies in the community could produce different results.

We respect the authors' alternative implementations of our idea, which adds to the healthy discussion of pure data-driven vs. interpretable hybrid models. Hopefully more research can go this way to understand the Pros and Cons of each method.

[Figure]

*Figure C1 Absolute percentage error between the observed peak discharge and the associated simulation value for the different models, classified by the return period of the observed peaks. The four categories to the right of the dashed vertical line present the errors associated with observed discharge above the 5-year return period threshold, evaluating the out-of-sample capabilities of the models. The n-value below each category indicates the amount of data used to produce the box-plot.*

[Figure]

*Figure C2. a) CDF of the 531 observed highest discharge values across all basins and their respective simulated values. The blue dots help visualize that under 3% of the events have values between 200 and 400 mm/day. b) Absolute percentage error of the 531 highest discharges for the different models.*

[Figure]

Figure C3. Comparisons in the experiments we have run, presented in Song et al, 2024.

| Reasons --- Impact | Differences | Comments |
|---|---|---|
| | | |

| | | |
|---|---|---|
| 1. Package difference

--- significant favor for LSTM | Espinoza24 used their own training framework built on NeuralHydrology (NH) package including a sequence-to-one LSTM network, while we employed our framework (HydroDL) with a sequence-to-sequence LSTM network with slightly different implementations. | While it seems the NH-hybrid package can give the same performance in the benchmark case used in Feng et al 2022, it gives suboptimal performance in extreme event tests. |
| 2. Ensemble strategy

--- large on NSE; maybe minor on extreme events | Espinoza24 used ensemble LSTM to compare with a single dHBV model | Espinoza24 misinterpreted the dHBV hybrid model. The multicomponent in HBV is not an "ensemble" . Rather, it is similar to the hidden size in LSTM. The real ensemble of differentiable models should be composed of different model structures, e.g., HBV, SAC-SMA, PRMS. We have ongoing work that shows the true model ensemble provides better NSE. That being said, the impact on extreme impact needs more time to understand. |
| 3. Experiment design

--- moderately favored LSTM impact on extreme events | Espinoza24 et al. validated the model using holdout years within the training period --- they trained the model with CAMELS data from 1980-2014, and tested it using years within 1980-2014 that had extreme events and were held out during training.

We trained from 1995/10/01 to 2014/09/30, holding out water years with peak flows greater than a 5-year return period. We then tested models on a separate continuous time period, from 1980/10/01 to 1995/09/30. | We argue a true test should purely exist in continuous history or future years to avoid any kind of information leak. Even though Espinoza24 would say the data in some years is held out for test, during training LSTM still sees what the future looks like after the extreme events. Somehow this makes it a simpler task than purely predicting in untrained years.

We have run the tests, a pure extrapolation like what we did represents a harder case, and LSTM shows more disadvantages less favored under such a more real-world scenario. |

Reference:

Feng, D., Liu, J., Lawson, K. and Shen, C., 2022. Differentiable, learnable, regionalized process-based models with multiphysical outputs can approach state-of-the-art hydrologic prediction accuracy. *Water Resources Research*, *58*(10), p.e2022WR032404.
https://agupubs.onlinelibrary.wiley.com/doi/full/10.1029/2022WR032404

Song, Y., Sawadekar, K., Frame, J.M., Pan, M., Clark, M., Knoben, W.J., Wood, A.W., Patel, T. and Shen, C., 2024. Improving Physics-informed, Differentiable Hydrologic Models for Capturing Unseen Extreme Events. *Authorea Preprints*. https://essopenarchive.org/doi/full/10.22541/essoar.172304428.82707157

Frame, J.M., Kratzert, F., Klotz, D., Gauch, M., Shalev, G., Gilon, O., Qualls, L.M., Gupta, H.V. and Nearing, G.S., 2022. Deep learning rainfall–runoff predictions of extreme events. *Hydrology and Earth System Sciences*, *26*(13), pp.3377-3392.
https://hess.copernicus.org/articles/26/3377/2022/hess-26-3377-2022.html

---

## Community Comment (CC3)

The authors replied:

For the 25-50 and 100+, we did not indicate in our manuscript that the LSTM performed better. We indicated that both models performed similarly, and the differences can be explained by statistical noise. This is still the case with dHBV1.0. As we showed in Figure B1 of our manuscript (see figure below), different random initializations of the LSTM can create variation in the reported metric. We can see that for the second row of the Figure below, the LSTM can achieve a median values of 0.51 and 0.56 for the 25-50 and 100+ intervals, which are close to the 0.51 and 0.55 reported by the dHBV1.0 for these same cases.

To test the authors' argument that this was statistical noise, we further run more random seeds on the dHBV1.0 (dHBV1.1p was not retrained with more random seeds as we don't have enough time before the comment session closes), as shown in Figure CC1 ("hybrid" is what Espinoza24 trained while dHBV1.0 are ours). It turns out the authors' argument was not correct. In all of these random seeds, we see a steady outperformance of dHBV1.0 over LSTM for 25-50, 50-100 and of course 100+ cases. In fact we can certainly run a statistical analysis to verify the statistical significance with more random seeds. LSTM is better in the 5-25 than dHBV1.0 but about the same as dHBV1.1p (only one random seed). That case precisely shows that LSTM is better at cases close to what is has seen in training, and worse for those cases that it has not seen.

[Figure]

Figure CC1. We re-ran the experiment with more random seeds. "hybrid" is what Espinoza24 trained while dHBV1.0 & dHBV1.1p are trained by us. Each row is the result from a random seed. The random seeds used for dHBV1.0 were 111111, 22222, 33333, 44444, 55555. The 1.1p was trained using the same random seed due to time limitation.

It seems fair to say the "hybrid" model trained by the authors is not representative of the dHBV1.0 as in every random seed the dHBV1.0 had smaller errors --- there is not even one exception. The difference between them is due to the different training frameworks employed, as we explained in the first comment. We leave it to other readers to interpret the differences, but, from our reading, the authors "hybrid" would suggest LSTM tend to outperform while our figure would suggest dHBV tend to outperform for the extremes. It seems fair to say the community would be better served by involving at least dHBV1.0-hydroDL into the comparison to draw a more balanced conclusion accordingly.

The point about input scaler

The authors further argued that some minor differences in the setups caused the difference. The authors replied:

> The authors (us) are calculating the mean and standard deviation used to standardize the input data using the whole period (training and testing). In our case, we calculated the statistics using only the training years, to avoid information leaking. We believe this might be one of several other reasons for the different results.
>
> During training, they constructed the batches using information from the whole period (1980-2014), which they send to the model in the forcTuple list. This includes both training and testing years. Therefore, during training, for some elements of the batch, the model does a forward pass of information contained in the testing regime. The associated simulated values are not used to calculate the loss during the optimization; however, this strategy is indeed different from the strategy we used.

First, this was a minor scaler setup and there was no data leakage because precipitation as an input is supposedly known or can be assumed for the purpose of calculating the scaler. Second, we proposed an experiment which cleanly and easily separates out training and test (the temporal extrapolation case shown in our first comment) where the advantages of dHBV were more prominent in our test and we again encourage the authors to run a case like that with NH. At least, through the temporal extrapolation case we can see that the issue mentioned by the author does not have a noticeable impact. Third, one can make some further effort to have a cleaner scaler. We have taken 99 steps to get close to their exact setup and trust the authors can bridge the last 1 step.

In reality, we don't encounter scenarios where we know both the historical and future time series and test in the middle of the time series. How we use the model is like what is shown in Figure CC2b. In this experimental design, the model is trained on water years with lower return periods (blue line), while water years with higher return periods (green line) are held out from training. The model is then tested over a separate time span that includes both extreme and low flow events.

[Figure]

(a)

[Figure]

(b)

Figure CC2. Different experimental designs.

The final point about ensemble.
We still think it is unfair to compare ensemble LSTM with a single dHBV. The multicomponent in dHBV is like the hidden size in LSTM. Here is a simple criterion: an ensemble of n LSTM has n neural networks, whereas a dHBV has only one neural network. Because of the constraint imposed by HBV, it is not as random as LSTM so random seeds are not what one should do to get an ensemble for dHBV. More effort will be shown down the road on this topic.

Overall, this point is not highly relevant to the extreme discussion, here, now. Nevertheless, comparing ensemble LSTM with a single dHBV still feels like "bringing everything you've got" on the LSTM side while not doing much on the dHBV side.

Finally, we would like to say that whether this paper gets published or not is not our concern --- we just want to ensure the community gets the full picture and get a balanced view.

---

## Author Comment (AC1)

**Response to CC1: 'Comment on egusphere-2024-2147', Chaopeng Shen, 22 Aug 2024**

In this document we answer the questions and comments given by Chaopeng Shen. In summary, we consider that our implementation of the hybrid model and the nature of the experiments conducted in our experiment are appropriate and correct. Possible reasons for the differences with the results presented by Chaopeng Shen will be explained below. For clarity, we will answer each comment individually and explain why certain choices have been made in the preparation of the manuscript.

1. Package difference

Chaopeng indicates that:

> *Espinoza24 used their own training framework built on NeuralHydrology (NH) package including a sequence-to-one LSTM network, while we employed our framework (HydroDL) with a sequence-to-sequence LSTM network with slightly different implementations.*
>
> *While it seems the NH-hybrid package can give the same performance in the benchmark case used in Feng et al 2022, it gives suboptimal performance in extreme event tests.*

This is not correct. Our hybrid model implementation in NH is also trained using sequence-to-sequence in the same manner has it has been implemented in Feng et al. (2022).

2. Ensemble strategy

Chaopeng indicates that:

> *Espinoza24 used ensemble LSTM to compare with a single dHBV model*
>
> *Espinoza24 misinterpreted the dHBV hybrid model. The multicomponent in HBV is not an "ensemble". Rather, it is similar to the hidden size in LSTM. The real ensemble of differentiable models should be composed of different model structures, e.g., HBV, SAC-SMA, PRMS. We have ongoing work that shows the true model ensemble provides better NSE. That being said, the impact on extreme impact needs more time to understand.*

We do not agree with this argument. There is no scientific reason why an ensemble should consist of different model structures. There are multiple ways in which an ensemble can be accomplished, for example: training multiple models with different initializations (like we did with the stand-alone LSTM), propagating the uncertainty of the inputs throughout the model to get a probabilistic output, using multiple models in parallel and weighting their outputs, among other strategies. E.g. in operational weather forecast practice, ensemble forecasts are mainly created by perturbed initial conditions.

Therefore, there is no reason why *a "real ensemble of differentiable models should be composed of different model structures"*. Chaopeng Shen indicates that they have done work in which different model structures give an advantage, and we do not doubt this is the case. It is perfectly feasible that different model structures do increase the performance. However, this does not indicate that using the same structure is not considered a "real ensemble". Moreover, their work on hybrid model ensembles was not published at the moment we ran the experiments, so we could not use this information in this study.

3. Experiment design

Chaopeng Shen indicates that:

> *Espinoza24 et al. validated the model using holdout years within the training period--- they trained the model with CAMELS data from 1980-2014, and tested it using years within 1980-2014 that had extreme events and were held out during training.*
>
> *We trained from 1995/10/01 to 2014/09/30, holding out water years with peak flows greater than a 5-year return period. We then tested models on a separate continuous time period, from 1980/10/01 to 1995/09/30.*
>
> *We argue a true test should purely exist in continuous history or future years to avoid any kind of information leak. Even though Espinoza24 would say the data in some years is held out for test, during training LSTM still sees what the future looks like after the extreme events. Somehow this makes it a simpler task than purely predicting in untrained years. We have run the tests, a pure extrapolation like what we did represents a harder case, and LSTM shows more disadvantages less favored under such a more real-world scenario.*
>
> *Further, we observed that alternative choices (Table C1) in experimental design, which might align more closely with Frame et al., 2022, or represent more realistic tests, could yield greater advantages for dHBV HydroDL over LSTM (Figure C3). Specifically, the experiments in Espinoza24 may allow LSTM to operate more within an "interpolation" regime, whereas true experiments should challenge models in an "extrapolation" context.*

We do not agree with this argument. It is not true that "*a true test should purely exist in continuous history or future years to avoid any kind of information leak. Even though Espinoza24 would say the data in some years is held out for test, during training LSTM still sees what the future looks like after the extreme events. Somehow this makes it a simpler task than purely predicting in untrained years*"

We included a buffer period of one year between the training and testing years, therefore the LSTM does not see" what the future looks like after the extreme events", nor is there any kind of information leaking from testing to training. The training/testing split was done based on the probability of the data, giving different conditions and presenting unseen events to the model during testing. Illustrating this idea with an example, we do not see any reason why, if a model was trained using low flow years in 1999 and 2003, this would give an advantage when predicting an extreme event in 2001.

**Possible reasons for the differences in performance**

Looking at the code uploaded by the authors, we found two possible reasons for the differences in the results.

Data standardization

In the code provided by Chaopeng Shen (https://colab.research.google.com/drive/12xUvTu9NoGVdcRWqJvypy4DGg5jy5q9T#scrollTo=v%20 0JIEuFZjkxq), the statistics to standardize the input data that goes into the LSTM are calculated as follows:

```
stat_dict={}
for fid, forcing_item in enumerate(forcingLst) :
    if forcing_item in log_norm_cols:
        stat_dict[forcing_item] = cal_stat_gamma(forcing_train[:,:,fid])
    else:
        stat_dict[forcing_item] = cal_stat(forcing_train[:,:,fid])
```

The numpy array *forcing_train* has dimensions (531, 12418, 3), and there are not any NaN values.

[Figure]

The second dimension indicates that one has 12 418 days (period between 1980-2014). Therefore, the authors are calculating the mean and standard deviation used to standardize the input data using the whole period (training and testing). In our case, we calculated the statistics using only the training years, to avoid information leaking. We believe this might be one of several other reasons for the different results.

Training strategy

In the code provided by Chaopeng Shen, the discharges are masked depending if one is in training/testing mode. Therefore, no testing discharge information is used to calculate the loss during training.

```
for gageidx, gage in enumerate(np.array(selected_camels)):
    gage = str(gage).zfill(8)
    print('Masking gage ', gage)

    for yridx in range(len(train_dates[gage]['start_dates'])):
        try:
            startTime = train_dates[gage]['start_dates'][yridx]

            endTime = train_dates[gage]['end_dates'][yridx]

            index_start = AllTime.get_loc(startTime)
            index_end = AllTime.get_loc(endTime)+1

            # Only use the water year with peak flow <5-year return period
            target_train_training_masked[gageidx,index_start:index_end] = target_train[gageidx,index_start:index_end]
        except:
            print("train mask failed for", startTime,' ', endTime)

    for yridx in range(len(test_dates[gage]['start_dates'])):
        try:
            startTime = test_dates[gage]['start_dates'][yridx]

            endTime = test_dates[gage]['end_dates'][yridx]

            index_start = AllTime.get_loc(startTime)
            index_end = AllTime.get_loc(endTime)+1

            # Only use the water year with peak flow <5-year return period
            target_test_training_masked[gageidx,index_start:index_end] = target_train[gageidx,index_start:index_end]
        except:
            print("test mask failed for", startTime,' ', endTime)

with open(CAMELS_path + f'/training_file_camels_1980_2014', 'wb') as f:
    pickle.dump((forcing_train, target_train_training_masked, attr_train), f)

with open(CAMELS_path + f'/testing_file_camels_1980_2014', 'wb') as f:
    pickle.dump((forcing_train, target_test_training_masked, attr_train), f)
```

However, their training strategy does differ from ours and it also differs from how they are training the stand-alone LSTM (given that they are using Neural Hydrology for the last one).

During training, they constructed the batches using information from the whole period (1980-2014), which they send to the model in the *forcTuple* list. This includes both training and testing years. Therefore, during training, for some elements of the batch, the model does a forward pass of information contained in the testing regime. The associated simulated values are not used to calculate the loss during the optimization; however, this strategy is indeed different from the strategy we used.

```
forcTuple = [forcing_train,forcing_LSTM_norm]
trainedModel = train.trainModel(
    model,
    forcTuple,
    streamflow_trans,
    attribute_norm,
    lossFun,
    nEpoch=EPOCH,
    miniBatch=[BATCH_SIZE, RHO],
    saveEpoch=saveEPOCH,
    saveFolder=out,
    bufftime=BUFFTIME)
```

For our hybrid model and for the stand-alone LSTM, the batches that we used during training do not contain any values located in the testing period, neither inputs nor targets, and during training no forward pass of any testing value is done. We believe that this consistency in training both models in the similar manner is beneficial.

Reported differences

Lastly, we would like to highlight that in 4 out of the 5 cases we evaluated, the differences in the models do not change the conclusions we showed. In Figure C1 uploaded by Chaopeng Shen (which we copy below for clarity) we can see the following:

[Figure]

Source: https://doi.org/10.5194/egusphere-2024-2147-CC1

    a. In the 1-5 and 5-25, the ranking between the LSTM with NH-Hybrid and LSTM with dHBV1.0 stays the same.
    b. For the 25-50 and 100+, we did not indicate in our manuscript that the LSTM performed better. We indicated that both models performed similarly, and the differences can be

explained by statistical noise. This is still the case with dHBV1.0. As we showed in Figure B1 of our manuscript (see figure below), different random initializations of the LSTM can create variation in the reported metric. We can see that for the second row of the Figure below, the LSTM can achieve a median values of 0.51 and 0.56 for the 25-50 and 100+ intervals, which are close to the 0.51 and 0.55 reported by the dHBV1.0 for these same cases.

[Figure]

**Figure B1.** Variation in absolute percentage error due to random initialization of the LSTM model.

c. In the 50-100 interval, it does looks like there is a difference between both implementations of the hybrid model (NH-Hybrid and dHBV1.0), that we believe can be explained by the reasons stated in the previous sections.

Sincerely,

Eduardo Acuna Espinoza on behalf of all co-authors

---

## Author Comment (AC2)

**Reply to CC2 of hess-2024-2147 (John Ding)**

Dear John Ding,

Thank you for your comment on our manuscript hess-2024-2147. Please find your comment below in blue, and our reply in black.

AR2 second-order autoregressive process of the streamflow

Besides the LSTM, HBV and a hybrid of the two, the authors may wish to revisit an autoregressive baseline model called AR(2) or AR2. This, an acceleration-based metric, is expressed by:

Qar2[t+1]=2Qobs[t]-Qobs[t-1], see Azmi et al. (2021, SC1, Eq. 1).

The subject was previously discussed between me and Uwe Ehret, the current closing author, on a storm event scale in a different but related context (ibid., AC1, Table 1). To summarize my take of our discussion, below are two main points:

1) a third-order AR model, AR-3 (Model-07, therein) when rounding off the time lag coefficients, is identical to AR2, and

2) it outperforms an ANN model (Model-08) by an NSE value of 0.99 to 0.12.

For the 531 CAMELS-US basins (Lines 125-130, and Figure 2), can we infer from point 2 above that an AR2 will be a better performing model? Let's consider this a hypothesis for falsification in another open discussion forum. In theory, an AR2 projection hydrograph over/under shoots the observed peak/trough flows - just visualize a USDA-SCS triangular unit hydrograph having an upslope and a downslope projection. This is in contrast to the authors' finding that 'all [three of their] models underestimated extreme flow scenarios,' (Line 243).

References

Azmi, E., Ehret, U., Weijs, S. V., Ruddell, B. L., and Perdigão, R. A. P.: Technical note: "Bit by bit": a practical and general approach for evaluating model computational complexity vs. model performance, Hydrol. Earth Syst. Sci., 25, 1103–1115, https://doi.org/10.5194/hess-25-1103-2021, 2021.

From the comment, we assume the John Ding suggests we should add an AR2 model to the set of models applied and compared in the manuscript. Although AR-models are known to perform well for streamflow time series due to their high temporal autocorrelation, we do not see merit in adding this model to the study, as it does not contribute to the goal of the paper, which is to investigate how well hybrid combinations of conceptual and LSTM-based hydrological models do in extrapolation towards very high streamflow, and to compare this to the two natural candidates for benchmark models, i.e. conceptual-only and LSTM-only. We therefore prefer to not include an AR-model into our work.

John Ding also raises the question " can we infer from point 2 above that an AR2 will be a better performing model?", with reference to Azmi et al. (2021). Respectfully we think this question is unrelated to the manuscript under review, and therefore invite the author to contact us outside the discussion forum of the manuscript to discuss this matter.

Sincerely,

Eduardo Acuna Espinoza on behalf of all co-authors

---

## Author Comment (AC3)

**General remark about benchmark models and comparability of our results to other model applications**

In our original publication, before running any extrapolation experiments, we benchmarked our model implementation against the study from Feng et al, (2022), which was the publicly available model at the time. We can see in Fig A1 of our original manuscript (which we repeat below for clarity), that the models' performance under the original conditions is almost identical.

[Figure]

Figure A1. Comparison of model performance against benchmark model – Feng et at, (2022).

Only after our model implementation was verified, we ran the extrapolation experiments, with exactly the same model characteristics we used to run our benchmarks. To our knowledge, we followed the best practices to ensure a fair and reproducible model comparison based on benchmark models reported in the scientific literature.

We would also like to stress that while we appreciate the efforts to compare our results to new model setups (dHBV1.1), the architecture we will refer to in our manuscript is the one proposed by Feng et al. (2022), due to the reasons stated above.

**About biased interests**

We want to clarify that we do not have any interest in favoring one model over the other, nor we want to "bring everything we got on the LSTM side while not doing much on the dHBV side" as suggested by Chaopeng Shen. For the LSTM we used different initializations, acting as an ensemble, because multiple publications ([2], [3], [4]) have shown that this technique produces more robust results. For the Hybrid model, we used a single initialization, because

that was the implementation done by Feng et at (2022), which is the one we were using as a reference.

**About the differences between the models (First comment by Chaopeng Shen in CC3)**

We appreciate that Chaopeng Shen ran additional experiments to test the influence on random seeds on dHBV1.0 performance. We also ran our hybrid model using multiple seeds, results we show in Fig B1. For higher return periods, in some cases the hybrid model performs slightly better, for other cases the LSTM performs slightly better. This is in accordance with what we expressed in the original text (Line 160-162), so there is no need for changing our original conclusions. We will include the updated figures and the updated results in a revised version of the manuscript.

[Figure]

Figure B1 (updated). Variation in absolute percentage error due to random initialization of the models (LSTM and Hybrid)

Chaopeng Shen indicates in his community comment, egusphere-2024-2147 CC3, that there are differences in the results he reported in his community comment, and the results we are reporting in our manuscript. We further analyzed these differences.

We compared the median Absolute Percentage Error from our results (Figure B1, shown above) with those from Chaopeng Shen's (Figure CC1, referenced in comment egusphere-2024-2147). The differences between the two sets of values were calculated and summarized in the table below

| Median Abs-Percentage Error | | | |
|---|---|---|---|
| Return period | NH-Implementation | Chaopeng Shen´s Implementation | Error |
| 5-25 | 0.46 | 0.45 | 2.2% |
| 25-50 | 0.53 | 0.51 | 3.8% |
| 50-100 | 0.57 | 0.54 | 5.3% |
| 100+ | 0.58 | 0.55 | 5.2% |
| Average | | | 4.1% |

We can see that the average difference is 4.1%, with a maximum difference of 5.3 %. We argue that these differences are small, especially if we consider the variation of the metric with the different random seeds. Reasons for the possible differences were discussed in our previous response. Therefore, we do not consider it necessary to modify the conclusions stated in our manuscript.

About the different experiment designs (comment "the point about input scaler" in CC3 by Chaopeng Shen)

In the community comment Chaopeng Shen shows an alternative experimental design, in which he does an additional temporal split of the data. He indicates that this experimental design cleanly separates the training and testing data.

We argue that our experimental design also does a clear and clean separation of the training and testing data. As explained in the manuscript, the training and testing conditions are clearly different, because we separate the water years based on the return period, and separate the two regimes by a buffer period as long as the model's sequence length, so absolutely no data

leakage from testing to training is possible. Additionally, by not doing an extra temporal separation, we are able to select training and testing years from the whole data record (1980-2014), which increases the size of our training and testing sets, and help us produce robust results.

Chaopeng Shen also indicates, regarding our training/test split, that "we don't encounter scenarios where we know both the historical and future time series and test in the middle of the time series". However, this never was the idea behind the setup from Frame et al. (2022). Indeed, as we have emphasized multiple times in our manuscript, the experimental design was intended as a stress-test to gain insight into the model's behavior during extreme events. For an operational model, one would not hold out any years, especially those containing extreme events.

About the comment "the final point about ensemble" in CC3 by Shaopeng Shen

About different interpretations of what defines an ensemble prediction, please see our reply 2 in AC2. About the suggestion by Chaopeng Shen that we "bring everything we've got on the LSTM side while not doing much on the dHBV side.", please see our above comment "about biased interests".

Sincerely,

Eduardo Acuna, on behalf of all co-authors

References

1. Feng, D., Liu, J., Lawson, K., & Shen, C. (2022). Differentiable, learnable, regionalized process-based models with multiphysical outputs can approach state-of-the-art hydrologic prediction accuracy. Water Resources Research, 58, e2022WR032404. https://doi.org/10.1029/2022WR032404

2. Gauch, M., Kratzert, F., Klotz, D., Nearing, G., Lin, J., & Hochreiter, S. (2021). Rainfall-runoff prediction at multiple timescales with a single Long Short-Term Memory network. *Hydrology and Earth System Sciences, 25*(4), 2045–2062. https://doi.org/10.5194/hess-25-2045-2021

3. Kratzert, F., Klotz, D., Shalev, G., Klambauer, G., Hochreiter, S., & Nearing, G. (2019). Towards learning universal, regional, and local hydrological behaviors via machine learning applied to large-sample datasets. *Hydrology and Earth System Sciences, 23*(12), 5089–5110. https://doi.org/10.5194/hess-23-5089-2019

4. Lees, T., Buechel, M., Anderson, B., Slater, L., Reece, S., Coxon, G., & Dadson, S. J. (2021). Benchmarking data-driven rainfall-runoff models in Great Britain: A comparison of long short-term memory (LSTM)-based models with four lumped conceptual models. Hydrology and Earth System Sciences, 25(10), 5517–5534. https://doi.org/10.5194/hess-25-5517-2021

---

## Author Comment (AC4)

**Response to RC1: 'Comment on egusphere-2024-2147', Basil Kraft**

We want to thank the referee for the detailed evaluation of our paper. In this document we answer the questions, comments and suggestions given. We will address those comments individually. For clarity, the original comments posted by the referee are written in blue.

**Short summary and highlights**

This study examines how well an established hybrid hydrological model generalizes under extreme conditions, comparing its performance to a data-driven model (LSTM) and a conceptual hydrological model. They split the discharge time series into training and test sets based on the magnitude of events within a hydrological year. Normal years were used for training, while extreme ones were used for testing.

The results indicate that the LSTM model performs best across all test data. For extreme events, the performance gap between the LSTM and hybrid models narrows, and eventually disappears for events with a 50-year return period or above. All models face challenges with extremely large discharge events, and the LSTM shows limitations due to saturation, where it cannot produce outputs beyond a certain threshold. While the hybrid and conceptual models slightly better reproduce extreme event patterns, they all exhibit substantial errors under these conditions.

Overall, the study suggests that while hybrid models may offer a modest advantage for modeling extreme discharge events, the LSTM performs better across all conditions. Thus, the choice of the best model should be guided by the specific objectives, such as whether the focus is on overall discharge or extreme conditions, as well as considerations for control and interpretability.

The study offers a critical perspective on hybrid models, which have gained popularity in recent years. While the benefits of hybrid models are often promoted, a thorough evaluation is still needed. Therefore, I find the study highly relevant. The study design is clear, and the use of established models lends credibility to the findings.

We thank the referee for the well-structured summary of our paper.

**Major remarks**

1. Could you explain why a fixed number of epochs was chosen for training instead of early stopping?

Response: The choice of using a fixed number of epochs instead of early stopping was done to keep consistency with the two studies we used as a base for our experiments ([1] and [2]). Moreover, early stopping based on validation scores requires a third split, a validation split, which means that we lose additional test data.

2. The study evaluates a specific hybrid model. While this is valuable, it is important to note that these findings may not apply to all hybrid models. I suggest updating the title to "Analyzing the

generalization capabilities of a hybrid hydrological model for extrapolation to extreme events",
and to discuss this limitation more in-depth.

Response: Thank you for the suggestion. We will modify the title accordingly. We did discuss the limitations in the original manuscript. In section 3.5, between lines 223-226, we stated that:

> "The comparison that we presented here was done using the hybrid architecture proposed by Feng et al. (2022). This architecture was chosen because it gave a competitive performance with LSTM in their original experiment and because the code was open source. Other hybrid model architectures might give different results, and we encourage the hydrological community to expand the test cases presented here."

However, we do admit that this is perhaps a bit spartan and will expand upon it in the revised manuscript.

3. If feasible, a comparison regarding low flow conditions would be appreciated.

Response: Thank you for the suggestion. Even though a comparison regarding low flow conditions would be interesting, our whole study and the proposed methodology focus on high flow conditions. Consequently, evaluating low flow conditions are outside of the scope of the current paper.

4. The overall structure could be improved. Currently, the Results section includes some Method descriptions, and within the Results and Discussion section, results and discussion are not clearly separated into paragraphs.

Response: Thank you for the suggestion. In a revised version of the manuscript, we will clearly separate the Results from the Method. However, with regard to the separation of Results and Discussion, we would like to keep these sections together, as we believe this creates a concise and fluent manuscript, and allows the reader to better understand the narrative of the study.

5. Although it may be beyond the scope of this study, examining the robustness of the (interpretable) hybrid model parameters would be very interesting.

Response: Thank you for the suggestion. In the revised version of the manuscript, we will add an Appendix discussing the variation of the dynamic parameters of the hybrid model.

6. The discussion falls a bit short in general. I would appreciate a more in-depth discussion of the findings.

Response: Thank you for the suggestion. In a revised version of the manuscript, we will provide a brief discussion of other advantages provided by the hybrid models, and some hypotheses about how our findings might generalize to other models.

**Minor remarks**

L11 Clarify "the latter study"

Response: We will modify this.

L9-L13 Consider revising the first paragraph for improved clarity.

Response: We will do that.

L35/L37 Choose either "large events" or "extreme events" for consistency.

Response: Thank you for pointing this out, we will use only the term "extreme".

L36 Perhaps rephrase to "How does a hybrid model compare to a process-based model"? See Major remarks.

Response: We will modify this in a revised version of the manuscript.

L38 Specify the type of advantage being discussed.

Response: We will modify this in a revised version of the manuscript.

L41 Rephrase to: "In Section 3, we compare the results of various tests that assess generalization capabilities."

Response: We will modify this in a revised version of the manuscript.

L50 A different name for the hybrid model might be clearer; "δ(γt, βt)" is somewhat cumbersome.

Response: This was done to maintain consistency with Feng et al., (2022), so we prefer keeping this name.

L53 Consider using "Experimental setup" as the section title.

Response: "Experimental setup" is a more general term. We would like to keep the title as "Data handling: training/test split" because it is more specific of what the subsection is about.

L60 Move "(a water year is defined as the period from October 1 to September 30)" to where you first mention" water years" on line 59.

Response: We will modify this in a revised version of the manuscript.

L71 Provide a reference for USGS NWIS.

Response: We will add the reference in a revised version of the manuscript.

L74ff Please rephrase for clarity.

Response: We will modify this in a revised version of the manuscript.

L89 Add a comma before "respectively".

Response: We will modify this in a revised version of the manuscript.

L89 Define NSE and provide a reference.

Response: We will modify this in a revised version of the manuscript.

L90 Specify which study is referred to as "the original study".

Response: We will modify this in a revised version of the manuscript.

L91 Provide full names and add a space between variable names and units.

Response: We will modify this in a revised version of the manuscript.

L92 Include categories of static variables, such as "27 static variables describing topography, soil properties, and land surface cover . . . ".

Response: We will modify this in a revised version of the manuscript.

L96 Briefly mention the benefits of using ensemble methods.

Response: We will add a short sentence, about the benefits of using ensembles, in a revised version of the manuscript.

L98 Replace "As mentioned in the introduction," with "For the hybrid model architecture,"

Response: We will modify this in a revised version of the manuscript.

L103 Rephrase to: "210 parameters (16 ensemble members, each with 13 HBV parameters plus 2 routing parameters)".

Response: We will modify this in a revised version of the manuscript.

L105 Did you use the same warm-up period for the LSTM, considering it also needs to initialize its states?

Response: The warmup period of the stand-alone LSTM is done during the 365 timesteps considered in the sequence length, so the warmup periods are equivalent. The main difference is that, with the LSTM, we only retrieved one value after the warmup period (seq-one) while in the hybrid we retrieved a whole year (seq-seq).

L107 Change "365 elements" to "365 time steps".

Response: We will modify this in a revised version of the manuscript.

L108 Provide the full name of the YAML file.

Response: We will modify this in a revised version of the manuscript.

L113 Rephrase: "To ensure a comprehensive comparison of the model spectrum,"

Response: We will modify this in a revised version of the manuscript.

L114 Rephrase to "14 parameters (12 HBV plus 2 routing)".

Response: We will modify this in a revised version of the manuscript.

L115 Note that this HBV instance has 12 parameters, while the hybrid model has 13.

Response: Yes. We explain the reason for this difference between lines 115-117.

L122 Rephrase to: "their generalization capabilities in the time domain to extreme events."

Response: We will modify this in a revised version of the manuscript.

L124 Consider "Model comparison for the whole test period" as a section title.

Response: We would like to keep the word "performance" because it is more specific of what the subsection is about.

L125 Introduce NSE before line 89.

Response: Thank you for pointing this out. We will modify this in a revised version of the manuscript.

L125 Change "reported for each model".

Response: We will modify this in a revised version of the manuscript.

L126 Rephrase: "The LSTM outperforms the hybrid model, with a median NSE of 0.75 and 0.71, respectively."

Response: We will modify this in a revised version of the manuscript.

L127 Rephrase: "The hybrid model has a median NSE of 0.64. This indicates that even with a different training-test split than the usual temporally contiguous subsets, our results align with those reported by Feng et al. (2022) and Acuña Espinoza et al. (2024), where the same model ranking was observed."

Response: Thank you for the suggestion. We will modify this in a revised version of the manuscript.

Fig. 2 Caption: "Cumulative Density Functions (CDF)"

Response: We will add the (CDF) abbreviation to the figure´s caption.

L131 Consider "Model Comparison for peak flows" as a section title.

Response: We would like to keep the word "performance" because it is more specific of what the subsection is about.

L135ff This section sounds more like methods; consider restructuring.

Response: Thank you for the suggestion. We will move this part to the method´s section.

L148 Omit: "This trend will also be observed in other experiments discussed in the following sections."

Response: We will remove this sentence from a revised version of the manuscript.

L150 The sentence is quite generic. Consider removing it and discussing the point in detail later.

Response: We would like to keep this sentence because it is supported by the results we showed in that section.

Fig. 3 Add ')': "The results of subplot b) show the error distribution."

Response: We will make the respective change.

L168 Use "Eq. 1" instead of "equation 1".

Response: We will modify this in a revised version of the manuscript.

L176 Replace "has to go out". Maybe "has to leave the system"? Also, the water could just stay in the system and accumulate over time, right?

Response: We will modify this in a revised version of the manuscript. About the second comment, your are correct, in the last time step there can still be water stored in the system. We will make the respective changes.

L181 Omit: "This phenomenon as such is not necessarily an indication of model deficiencies."

Response: We would like to keep this sentence because it helps us connect with the next idea.

L228 Replace "For the reasons stated in previous sections, . . . " with "This stress-testing . . . "

Response: We would like to keep the sentence structure as it is, because it helps us connect with the next idea.

L229 "In a practical case, one should use all the data during model training, to increase the performance of the models." Could you mention that you mean using also extreme events, and not literally all data (because we want training/test split)?

Response: We will modify this in a revised version of the manuscript.

L259 Replace "As explained in the manuscript," with "For our hybrid model,"

Response: We will modify this in a revised version of the manuscript.

L260 Please rephrase: "Because our experiment pipeline was executed in the NeuralHydrology package, we did first had to benchmark our model . . . ".

Response: We will modify this in a revised version of the manuscript.

Final remarks

We would like to thank the referee for the overall positive evaluation of our manuscript and hope we could address the questions raised in a satisfactory manner.

References

1. Feng, D., Liu, J., Lawson, K., & Shen, C. (2022). Differentiable, learnable, regionalized process-based models with multiphysical outputs can approach state-of-the-art hydrologic prediction accuracy. Water Resources Research, 58, e2022WR032404. https://doi.org/10.1029/2022WR032404
2. Frame, J. M., Kratzert, F., Klotz, D., Gauch, M., Shalev, G., Gilon, O., Qualls, L. M., Gupta, H. V., & Nearing, G. S. (2022). Deep learning rainfall-runoff predictions of extreme events. *Hydrology and Earth System Sciences, 26*(13), 3377–3392. https://doi.org/10.5194/hess-26-3377-2022

---

## Author Comment (AC5)

**Response to RC2: 'Comment on egusphere-2024-2147', Shijie Jiang**

We want to thank the referee for the detailed evaluation of our paper. In this document we answer the questions, comments and suggestions given. We will address those comments individually. For clarity, the original comments posted by the referee are written in blue.

The manuscript "Analyzing the generalization capabilities of hybrid hydrological models for extrapolation to extreme events" compares the generalization capabilities of hybrid models, LSTM networks, and process-based models for rainfall-runoff simulations, with a particular focus on extreme events. The study examines whether hybrid models provide a meaningful advantage over standalone data-driven or process-based models. The results suggest that hybrid models show marginal improvements in predicting extreme peak flows, but overall perform similarly to LSTM networks. The authors argue that given the comparable performance, the choice of model depends on user needs. Overall, the study does a great job of providing a balanced perspective on the hybrid models. The paper is valuable in stimulating further discussion in the field.

We thank the referee for the well-structured summary of our paper.

Major comments

1) One of the central claims for hybrid models is that they combine the predictive power of data-driven approaches with the interpretability of process-based models. However, the manuscript focuses more on marginal differences in predictive performance than on the added interpretability that might justify hybrid models. I suggest including a discussion of the trade-off between accuracy and interpretability. For example, does the hybrid model help to better understand the causes of extreme flows, such as snowmelt, soil moisture dynamics, or precipitation anomalies? Could the explicit encoding of hydrologic concepts in the hybrid model be more valuable for decision making, even if the predictive gains are minimal?

Response: Thank you for the suggestion. We agree that both performance and interpretability can be important, and that is why in previous studies, as Acuña Espinoza et al. (2024), we ran multiple experiments to evaluate model interpretability. In this study, we are tackling the question from a practitioner's point of view, in which performance is the main interest. However, in the Limitation section, we will add more emphasis on the fact that other criteria, besides model accuracy, play an important role in an integral evaluation of the model.

2) While the paper touches on model errors during extreme events, it does not provide an analysis of where and why each model is better or worse, e.g., under which geophysical, climatic, or soil conditions. This could be helpful to better understand the strengths and limitations of each model type and provide a useful guide to when hybrid / LSTM models are most beneficial.

Response: Thank you for the suggestion. In a revised version of the manuscript we will include a map, indicating for each basin, the difference in performance of the models. This way we will be able to visualize if there are geographic settings in which one model consistently outperforms the other.

3) A related comment is that while the authors conclude that the choice of model depends on user needs, the manuscript does not provide clear guidance on how to make this choice. For example, in data-poor environments where high-quality or long-term observational data may not be available, should hybrid models be preferred because they incorporate process-based knowledge that could compensate for sparse data? Is it possible to make a comparison that assumes limited data? I think it would be helpful for practitioners working in regions with poor monitoring infrastructure.

Response: Thank you for the suggestion. To evaluate if hybrid models can be trained with less data is by itself a full study, with a different set of experiments, which is beyond the scope of our study. A good overview on related studies for process- and data-based hydrological models is given by Jiang et al. (2024). Specifically for the fully integrated hydrological model (ATS), they conclude that about 4 years of data allow for robust parameter learning. From our own recent work comparing the learning ability of single-basin process-based (HBV) and data-based (LSTM) models (publication in preparation), we found that HBV learns all it can from 2-3 years of data, that the LSTM achieves good performance with 2-3 years of training data but keeps learning when more data are available, and that LSTM outperforms HBV beyond ca. 10 years of training data.

Moreover, both data-driven methods and hybrid models have shown to perform better when trained regionally. Therefore, even in data-sparse regions, one could train the models on public databases and then fine-tune it to the specific areas, which would somehow help mitigate the limited data problem. Therefore, even though this is an interesting question, we believe it is considerably outside of the scope of the current study. Nevertheless, we will include a short discussion, similar to the above paragraph, in the revised manuscript.

Specific comments:

L12, the term "out-of-sample conditions" is somewhat ambiguous. Please specify what type of generalization is meant (temporal or spatial domains).

Response: Agreed. We will better specify this term in a revised version of the manuscript.

L16, the phrase "notion of interpretability" could be clearer. What does "notion" mean in this context? It sounds vague. If interpretability is considered to be a key reason for adopting hybrid models over purely data-driven ones, it should be more clearly defined and quantified. Does interpretability mean the ability to interpret the parameters, processes, or outputs in a hydrologically meaningful way? Or are you suggesting that it's a "so-called" interpretability?

Response: The interpretability gained in this type of hybrid model is that we associate the parameters and buckets of the process-based models with interpretable processes, domains and states (baseflow, interflow, snow accumulation…). However, we argue that this type of interpretability is based on association, and the physical principles represented on process-based models, such as the HBV, have major simplifications. We will clarify this in a revised version of the manuscript.

L30, what specific structural deficiencies are you referring to here?

Response: The hybrid structure we present in this study consists of a data-driven part that predicts the parameters (static and time-varying) used to operate a process-based model. This is the same architecture type used in Acuña Espinoza et al. (2024). They show that the data-driven part, through the dynamic parameterization, is able to increase the performance of the model, compared to the stand-alone process-based benchmarks. This was attributed to the fact that process-based models present a relatively simple structure that in a lot of cases oversimplifies the actual physical processes. One example is assuming that all the flows have a linear relationship with the storage and that the storage/discharge rate does not change over time. Or that snow melting is a linear process, proportional to the difference between a threshold temperature and the air temperature.

By giving additional flexibility through the dynamic parameterization, the LSTM is able to compensate for some of these deficiencies. Acuña Espinoza et al. (2024) discussed these aspects in more detail, and that is why we refer to that publication. We believe that including that in our current manuscript would reduce the fluency of the reading, as it is not the main point we are trying to establish.

L35, the focus on "higher predictive accuracy" may overlook the fact that accuracy alone may not be the best criterion for assessing model suitability. Authors should clarify that other criteria (such as robustness, model transparency, applicability) besides accuracy may be equally important in model evaluation.

Response: We agree that other criteria, besides model accuracy, play an important role in an integral evaluation of the models. However, in this study, the main focus of our experiments is model accuracy. In a revised version of the manuscript we will indicate that while we focus on accuracy in this study, future studies can expand the comparison tests in the other points.

L100, the explanation of the hybrid model's parameterization is complex and may not be easily understood by just reading this paper. At least a clearer explanation of the buckets and parameters is needed.

Response: Thank you for the suggestion. In a revised version of the manuscript we will add a figure of the model setup in an Appendix, to better illustrate the idea.

L127 without discussing the potential limitations of the HBV model, this claim seems overly simplistic. It is useful to explain here why the HBV model underperformed, even though it has been studied in previous studies.

Response: In a revised version of the manuscript we will add some explanation of why the HBV model has a lower performance. Something similar to what we indicated in our previous response to the referee's comment on L30.

L150, again, this conclusion of equivalence is overly simplistic and could lead to believing that there are no meaningful differences between the models. Are there certain types of basins or hydrological conditions (e.g., arid basins) where one model clearly outperforms the other?

Response: As discussed in Major Comment 2, we will add a map to indicate if under certain conditions, one model outperforms the other.

L167, it's hard to read from the figure about the "slightly lower errors".

Response: In a revised version of the manuscript, we will update Figure B1 with additional runs for the hybrid model using different seeds, which will give us more information about the differences between the models.

L215, this observation is important but lacks sufficient follow-up. If the dynamic parameterization reaches its limits during extreme events, it indicates a potential flaw in the model design, but the text does not discuss how this issue could be addressed or what its implications are. Could the predefined intervals be adjusted or extended to better handle extreme events?

Response: We defined the parameter intervals according to Feng et al, (2022), which was the model we were using as a benchmark. In a revised version of the manuscript we will expand on strategies to address these implications.

L220, I am very confused here. How does the snowmelt effect indicate the potential bias in the input data? If the snowmelt flux is high, it's not surprising to see a discrepancy between precipitation and runoff. This statement also raises the question of a structural flaw in the HBV model, but it is not elaborated. I'm left wondering what specific deficiencies in the snow module are responsible for the poor performance and how these deficiencies could be addressed in future work. For example, is the snowmelt process not adequately modeled due to insufficient temperature data, or is the parameterization of the snow module too simplistic?

Response: In this paragraph we are looking at possible causes of why the hybrid model underestimates the peak discharges. We mentioned that for the specific events presented for basins 14182500 and 14138800, the cumulative water volume that comes from precipitation is smaller than the cumulative water volume given by the observed discharge. Given the mass conservative structure of the hybrid model, the simulated values will therefore be smaller than the observed discharge, unless most of the simulated discharge comes from snowmelt. However, given the flow underestimation, this is not the case. In a revised version of our manuscript, we will make the explanation more understandable.

We also agree with the referee that two possible deficiencies in the snow module could be insufficient temperature data and an overly simplistic parameterization of the snow module. However, because we are conducting a regional study in 531 basins, looking in detail at model deficiencies in each basin is not feasible, nor is it the main point of our study.

L225, it's vague and doesn't provide enough insight into what types of hybrid architectures might yield different results. In my opinion, the hybrid model used in this paper considers model with a conceptual model as the backbone and neural networks for parameter learning. It would be more actionable to point out some other types of hybrid models, e.g., component replacement or more conceptual frameworks (e.g., https://hess.copernicus.org/articles/26/1579/2022/) that might address some of the limitations identified in the study.

Response: In a revised version of the manuscript we will expand upon this point, and add the reference provided by the referee.

L230, I'm afraid this recommendation is too general and simplistic…

Response: If we understood correctly, the recommendation the referee refers to is: "In a practical case, one should use all the data during model training, to increase the performance of the models." We would argue that this recommendation is correct, and it was previously stated in Nevo et al (2022).

L241, is it possible to use more precise numbers or statistical analysis to support the claim of "slight" outperformance. If the differences are marginal, do you think they might still matter in practical scenarios?

Response: We did additional runs for the hybrid model using different seeds. In a revised version of the manuscript we will report those results, which will provide more information.

L245, the mention of "possible bias in the input data" is speculative without further analysis. And if that's the case, does it imply that LSTM is insensitive to the bias?

Response: In the analysis accompanying Figure 6 of our manuscript, we point to precipitation bias as a possible cause for the peak underestimation in the hybrid models. Similar discussions have been carried out in the literature, indicating that biases in precipitation measurements can be caused by point uncertainty, interpolation uncertainty, and equipment malfunction (Westerberg & McMillan, 2015; Bárdossy & Anwar, 2023), especially if one is working with catchment-averaged values. Therefore, we believe that our hypothesis is correctly justified. Doing a bias analysis for the whole CAMELS-US dataset is outside of the scope of our current publication.

About the second question, the LSTM does not have a mass conservative structure, and therefore, systematic biases in the inputs can be accounted for. We will add this discussion and references to the revised manuscript.

L249, the statement about dynamic parameterization is not sufficiently elaborated. It doesn't provide enough detail about how this adaptation happens or why it is particularly useful for extreme events. Also, the comparison with LSTM gating is interesting, but lacks further discussion.

In a revised version of the manuscript we will expand upon this point.

**Final remarks**

We would like to thank the referee for the overall positive evaluation of our manuscript and hope we could address the questions raised in a satisfactory manner.

**References**

1. Acuña Espinoza, E., Loritz, R., Álvarez Chaves, M., Bäuerle, N., & Ehret, U. (2024). To bucket or not to bucket? Analyzing the performance and interpretability of hybrid hydrological models with dynamic parameterization. Hydrology and Earth System Sciences, 28(12), 2705–2719. https://doi.org/10.5194/hess-28-2705-2024

2. Bárdossy, A., & Anwar, F. (2023). Why do our rainfall–runoff models keep underestimating the peak flows? *Hydrology and Earth System Sciences, 27*(10), 1987–2000. https://doi.org/10.5194/hess-27-1987-2023

3. Feng, D., Liu, J., Lawson, K., & Shen, C. (2022). Differentiable, learnable, regionalized process-based models with multiphysical outputs can approach state-of-the-art hydrologic prediction accuracy. Water Resources Research, 58, e2022WR032404. https://doi.org/10.1029/2022WR032404

4. Nevo, S., Morin, E., Gerzi Rosenthal, A., Metzger, A., Barshai, C., Weitzner, D., Voloshin, D., Kratzert, F., Elidan, G., Dror, G., Begelman, G., Nearing, G., Shalev, G., Noga, H., Shavitt, I., Yuklea, L., Royz, M., Giladi, N., Peled Levi, N., Reich, O., Gilon, O., Maor, R., Timnat, S., Shechter, T., Anisimov, V., Gigi, Y., Levin, Y., Moshe, Z., Ben-Haim, Z., Hassidim, A., & Matias, Y. (2022). Flood forecasting with machine learning models in an operational framework. *Hydrology and Earth System Sciences, 26*(15), 4013–4032. https://doi.org/10.5194/hess-26-4013-2022

5. Westerberg, I. K., & McMillan, H. K. (2015). Uncertainty in hydrological signatures. *Hydrology and Earth System Sciences, 19*(9), 3951–3968. https://doi.org/10.5194/hess-19-3951-2015

6. Jiang, P., Shuai, P., Sun, A. Y., and Chen, X.: Optimizing parameter learning and calibration in an integrated hydrological model: Impact of observation length and information, Journal of Hydrology, 643, 131889, https://doi.org/10.1016/j.jhydrol.2024.131889, 2024.

---

## Author Response (AR1)

**Author´s response**

Dear Prof. Dr. Brunner

Please find attached with this document the new version of our manuscript, which incorporates the changes suggested during the review process. Attached to this document, you will find the revised manuscript along with a tracked changes version. Below, we provide a summary of the major changes made.

1. The content of the previous **Figure 3** has been incorporated into the former **Figures 1** and **4**. As a result, the original **Figure 3** has been removed, and the remaining figures have been renumbered.

2. To ensure a clearer separation between methods and results, we introduced a new section, **2.3 Performance Metrics**, which contains information previously included in the results section.

3. We added a new section, **3.4 Spatial Analysis of Model Performance**, expanding our analysis to include the comparison of model performance across space. This section includes new figure.

4. In **3.5 Saturation Analysis** and **3.6 Limitations and Uncertainties**, we expanded our discussions.

5. We added additional information to **Figure 5**.

6. Section **4. Summary and Conclusions** has been expanded.

7. Appendices:

   o **Appendix A** now includes a new figure explaining the structure of the hybrid model.

   o **Appendix B** now includes results from additional runs of the hybrid model, allowing for a more robust evaluation of model differences.

   o **Appendix C** includes supplementary figures related to the spatial analysis of model performance.

   o **Appendix D** discusses the temporal variation of the hybrid model's dynamic parameters in detail.

Even with these changes, the main points of the paper did not change. We believe the modifications made extensively cover the changes proposed by both referees. We would like to thank both referees, as their input in the review process allowed us to produce a better manuscript.

Kind regards,
Eduardo Acuña on behalf of the co-authors.

---

## Referee Report (RR1)

**"Analyzing the generalization capabilities of hybrid hydrological models for extrapolation to extreme events"**

**Manuscript #*2024-2147***
**Second iteration**
**B. Kraft**

December 9, 2024

**Minor remarks**

L31 "...does provide  interpretability ..."

L36 "Building on this research line and ..."

L44 "Does  **hybrid modeling** offer ..."

L61 "...we split the training and test  **set temporally** by years ..."

Figure 1 Use "Time" or "Date" as label for x-axis (also in Figure 5). The legend label "obs" should be changed to "Observed", also for consistency with Figure 5.

L151 " **This** is not surprising,  **as** ..."

L154 "...attributed to the fact that **conceptual** process-based models ...". Some process-based models are very complex, maybe use "conceptual"?

L158 "Moreover, we  show ..."

Figure 2 Here and in other figures: I usually re-introduce abbreviations (NSE, later APE) in figure captions.

225 "This saturation limit **could** explain ..."

Figure 6 For consistency, I suggest to remove the last sentence in the caption. You don't do this for figure 1, for example, and it is clear anyway.

L261 Also glacier melt in addition to snow?

L280 Semicolon between "Kraft et al. (2022); Hoge et al. (2022)" probably wrong.

S3.6 Limitations you could mention here taht you did not test for spatial generalizability.

Appendix D I appreciate the inclusion of a brief analysis of the learned hybrid model dynamic parameters in Appendix D. The appendix/figure is not mentioned in the main body. This would be an opportunity to briefly discuss "interpretability" of hybrid models and the usefullness for understanding / debugging / establishing trust in the model.

L393 It is **Högge** with umlaut.

---

## Author Response (AR2)

**Author´s response**

Dear Prof. Dr. Brunner

Please find attached with this document the new version of our manuscript, which incorporates the changes suggested during the review process. Attached to this document, you will find the revised manuscript along with a tracked changes version. Below, we provide a point-by-point answer to the changes proposed by Dr. Kraft, which are written in blue.

L31 ". . . does provide  of interpretability . . . "
Response: Thank you for the suggestion. We did the respective change.

L36 "Building on this research line and . . . "
Response: Thank you for the suggestion. We did the respective change.

L44 "Does  hybrid modeling offer . . . "
Response: Thank you for the suggestion. We did the respective change.

L61 ". . .we split the training and test  set temporally by years . . . "
Response: Thank you for the suggestion. We changed the word "periods" to "set".

Figure 1 Use "Time" or "Date" as label for x-axis (also in Figure 5). The legend label "obs" should be changed to "Observed", also for consistency with Figure 5.
Response: Thank you for the suggestion. We changed the x-axis label to "Date", and modified the legend to "observed".

L151 " This is not surprising, and as . . . "
Response: Even if the idea is repeated in two consecutive sentences, we believe it makes a clearer statement, so we would like to keep it as it is.

L154 ". . . attributed to the fact that **conceptual** process-based models . . . ". Some process-based models are very complex, maybe use "conceptual"?
Response: Thank you for the suggestion. We did the respective change.

L158 "Moreover, we  show . . . "
Response: Thank you for the suggestion. We did the respective change.

Figure 2 Here and in other figures: I usually re-introduce abbreviations (NSE, later APE) in figure captions.
Response: Thank you for the suggestion. We re-introduce the abbreviations in all the figures.

225 "This saturation limit **could** explains . . . "
Response: Thank you for the suggestion. We did the respective change.

Figure 6 For consistency, I suggest to remove the last sentence in the caption. You don't do this for figure 1, for example, and it is clear anyway.
Response: Thank you for the suggestion. We did the respective change.

L261 Also glacier melt in addition to snow?
Response: Thank you for the suggestion. We incorporated glacier melting into the text.

L280 Semicolon between "Kraft et al. (2022); Hoge et al. (2022)" probably wrong.
Response: Thank you for the suggestion. We changed the citation type.

S3.6 Limitations: you could mention here that you did not test for spatial generalizability.
Response: Thank you for the suggestion. We added this to the limitations section.

Appendix D I appreciate the inclusion of a brief analysis of the learned hybrid model dynamic parameters in Appendix D. The appendix/figure is not mentioned in the main body. This would be an opportunity to briefly discuss "interpretability" of hybrid models and the usefullness for understanding / debugging / establishing trust in the model.
Response: Thank you for the suggestion. In the first paragraph of the limitations, we included the reference to the appendix and expanded on the importance of model interpretability.

L393 It is **Högge** with umlaut.
Response: Thank you for pointing this out, we added the umlaut. About the double "g", we doubled checked the paper, and the last name only contains one "g".
* * *
We believe the modifications made cover the changes proposed by Dr Kraft. We would like to thank both referees, as their input in the review process allowed us to produce a better manuscript.

Kind regards,
Eduardo Acuña on behalf of the co-authors.